# XFacta: Contemporary, Real-World Dataset and Evaluation for Multimodal Misinformation Detection with Multimodal LLMs

## Abstract

The rapid spread of multimodal misinformation on social media calls for more effective and robust detection methods. Recent advances leveraging multimodal large language models (MLLMs) have shown the potential in this challenge. However, it remains unclear exactly where the bottleneck of existing approaches lies (evidence retrieval *v.s.* reasoning), hindering the further advances in this field. On the dataset side, existing benchmarks either contain *outdated* events, leading to evaluation bias due to discrepancies with contemporary social media scenarios as MLLMs can simply memorize these events, or artificially *synthetic*, failing to reflect real-world misinformation patterns. Additionally, it lacks comprehensive analyses of MLLM-based model design strategies. To address these issues, we introduce XFacta, a contemporary, real-world dataset that is better suited for evaluating MLLM-based detectors. We systematically evaluate various MLLM-based misinformation detection strategies, assessing models across different architectures and scales, as well as benchmarking against existing detection methods. Building on these analyses, we further enable a semi-automatic detection-in-the-loop framework that continuously updates XFacta with new content to maintain its contemporary relevance. Our analysis provides valuable insights and practices for advancing the field of multimodal misinformation detection.

## 1 Introduction

A lie can travel halfway around the world before the truth can get its boots on—a statement that feels especially true in the age of social media. As platforms enable information to spread rapidly, humans face increasing challenges in identifying fake news online. Modern fake news is often multimodal, combining text with images that appear to support false or unrelated events, which makes detection more challenging. The rise of deepfake technology further lowers the barrier to creating such deceptive content. These developments highlight the need for more advanced and robust methods to automatically detect multimodal misinformation.

The emergence of multimodal large language models (MLLMs), with strong reasoning capabilities across both text and images, offers a promising direction for detecting multimodal misinformation. Recent studies have begun to explore this potential. Some methods (Qi et al., 2024; Liu et al., 2024a; Zeng et al., 2024; Shalabi et al., 2024) fine-tune a general-purpose MLLM on specific misinformation datasets to create task-specific models. Other approaches (Khaliq et al., 2024; Xuan et al., 2024; Liu et al., 2024b; Geng et al., 2024) adopt a zero-shot setting and rely on more powerful models such as GPT-4 or Gemini, which achieve better performance on existing misinformation datasets. In general, the existing MLLM-based misinformation detectors mimic human verification processes, which involves two main steps: *evidence retrieval*, where external information is retrieved from Internet to serve as evidence, then *reasoning*, where the news post and the retrieved evidence are systematically analyzed and combined to make final judgment.

Despite the promising results reported in these studies, it remains unclear exactly where the bottleneck of existing MLLM-based misinformation detection methods lies (evidence retrieval *v.s.* reasoning), hindering further advances in this field. From the **dataset perspective**, misinformation on real-world social media often involves novel and timely events that are absent from MLLMs' train-

Figure 1: **Left**: An example from our dataset, where the MLLM (GPT-4o) must rely on evidence to judge real or fake. **Right**: An example from non-real-time datasets, where evidence matters less. Evaluating MLLM-based misinformation detectors on XFACTA introduces less evaluation bias.

ing data. Detecting these events requires models to actively retrieve evidence and reason thoroughly based on them. In contrast, existing misinformation benchmarks (Vlachos & Riedel, 2014; Wang, 2017; Thorne et al., 2018; Hanselowski et al., 2019; Khanam et al., 2021) contain mostly *outdated* data with events that may already exist in the training data of MLLMs, allowing models to rely simply on *memorization* rather than evidence-based reasoning. It introduces a significant evaluation bias as evidenced by an example shown in Fig. 1. In addition, some datasets (Luo et al., 2021; Chakraborty et al., 2023; Liu et al., 2024b; Shao et al., 2023; Aneja et al., 2021) are *synthetic*, meaning that misinformation samples are artificially constructed using AI models rather than collected from real-world sources. This limits their ability to reflect the complexity and strategies used by real misinformation creators. Regarding **technical approaches**, while existing studies typically focus on proposing new models or methods and demonstrating their effectiveness on specific datasets, it lacks *systematic analyses* and *rigorous comparisons* of different design choices for MLLM-based detection. Consequently, it still remains difficult today to identify best practices or generalizable insights for building reliable multimodal misinformation detectors.

In this paper, we address these limitations by curating a new misinformation dataset, named XFACTA (collected from **X** (Twitter) and for **Fact**-checking). All data points are from after January 2024, ensuring its *contemporary* relevance (e.g., more recent than the October 2023 cutoff of GPT-4o). Moreover, they are sourced from rumor spreaders on social media, reflecting patterns observed in the *real world*. Based on this dataset, we conduct a systematic exploration of how to build an MLLM-based misinformation detector from the perspectives of evidence retrieval and reasoning, respectively. Additionally, we evaluate various MLLMs of different architectures and scales, as well as existing misinformation detection approaches. From these experiments and analyses, we provide valuable insights on MLLM-based misinformation detection. Building on these insights, we apply the resulting detector to flag new posts with preliminary assessments for human reviewers to verify and add to XFACTA. This semi-automatic detection-in-the-loop cycle keeps the dataset up to date and prevents it from becoming outdated over time. We believe the XFACTA dataset and our study results will serve as a useful benchmark for future research in multimodal misinformation detection.

To conclude, our contributions are:

- We curate a contemporary, real-world dataset for multimodal misinformation detection and integrate a semi-automatic detection-in-the-loop process to keep it continuously up to date, which will further advance MLLM-based detection research.
- With XFACTA, we provide a comprehensive and in-depth analysis of developing a good MLLM-based misinformation detection model from two perspectives: evidence retrieval and reasoning, offering valuable insights to the field.
- We conduct a comprehensive evaluation of various MLLM-based misinformation detection strategies, assessing models across different architectures and scales, as well as benchmarking against existing detection methods.

## 2 RELATED WORK

**Datasets:** Previous studies have introduced various unimodal text-based misinformation datasets (Vlachos & Riedel, 2014; Wang, 2017; Thorne et al., 2018; Hanselowski et al., 2019;

Table 1: Comparison of different misinformation datasets. `Contemporary` refers to data published after January 1, 2024; `Real-world` means fake posts created by actual users, not artificially generated using AI models; and `Evidence-based annotations` mean there are annotations supported by sufficient evidence to verify the data.

| Dataset | Multimodal | Contemporary | Real-world | Evidence-based annotations | Real Num | Fake Num |
|---|---|---|---|---|---|---|
| FEVER (Thorne et al., 2018) | ✗ | ✗ | ✗ | ✓ | 93,367 | 43,107 |
| LIAR (Wang, 2017) | ✗ | ✗ | ✓ | ✓ | 7,085 | 5,751 |
| NewsCLIPpings (Luo et al., 2021) | ✓ | ✗ | ✗ | ✗ | 816,922 | 816,922 |
| Fakeddit (Nakamura et al., 2019) | ✓ | ✗ | ✓ | ✗ | 527,049 | 628,501 |
| Snopes+Reuters Zlatkova et al. (2019) | ✓ | ✗ | ✓ | ✓ | 592 | 641 |
| DGM$^4$ (Shao et al., 2023) | ✓ | ✗ | ✗ | ✗ | 77,426 | 152,574 |
| FACTIFY 3M (Chakraborty et al., 2023) | ✓ | ✗ | ✗ | ✗ | 406,000 | 316,000 |
| MMFakeBench (Liu et al., 2024b) | ✓ | ✗ | ✗ | ✗ | 3,300 | 7,700 |
| COSMOS (Aneja et al., 2021) | ✓ | ✗ | ✗ | ✓ | 1,700 | 1,700 |
| Mocheg (Yao et al., 2023) | ✓ | ✗ | ✓ | ✓ | 5,144 | 5,855 |
| MediaEval (Boididou et al., 2016) | ✓ | ✗ | ✓ | ✓ | 292 | 410 |
| VERITE (Papadopoulos et al., 2023) | ✓ | ✗ | ✓ | ✓ | 338 | 662 |
| Post-4V (Geng et al., 2024) | ✓ | ✓ | ✓ | ✓ | 81 | 105 |
| **XFACTA (Ours)** | ✓ | ✓ | ✓ | ✓ | 1,200 | 1,200 |

Khanam et al., 2021). The rise of social media has led to increasing attention on multimodal misinformation detection, along with the release of various datasets (Nakamura et al., 2019; Zlatkova et al., 2019; Yao et al., 2023; Boididou et al., 2016; Papadopoulos et al., 2023). However, real-world misinformation datasets are typically either small in size or suffer from noisy annotations. Therefore, some other works (Luo et al., 2021; Chakraborty et al., 2023; Liu et al., 2024b; Shao et al., 2023; Aneja et al., 2021) leverage heuristic rules or AI models to synthesize datasets for misinformation detection. As a consequence of such synthesis, these datasets fail to capture real-world misinformation creators' complex patterns and strategies. In addition, all of the above datasets are not contemporary and often overlap with the training data of MLLMs, which prevents a fair and robust evaluation of MLLM-based misinformation detectors. Post-4V (Geng et al., 2024) addresses this by collecting more recent examples, but its size is very limited, and data collection and processing details are underdocumented, making it less suitable as a widely accepted baseline. In contrast, our XFACTA dataset ensures both contemporary and real-world characteristics, while maintaining a moderate scale that is sufficient to evaluate MLLMs in a zero-shot setting. In addition, our dataset provides detailed journalist evidence for fake news, which can help validate the reasoning paths of detection models. A multi-dimensional comparison across different datasets can be found in Table 1.

**Models:** Some traditional multimodal misinformation detectors Abdelnabi et al. (2022); Yuan et al. (2023); Brahma et al. (2023a); Aneja et al. (2023); Mu et al. (2023); Zhang et al. (2023); Brahma et al. (2023b); Yang et al. (2024) are trained and evaluated on specific datasets, such as the commonly used NewsCLIPpings dataset (Luo et al., 2021). With the emergence of open-source MLLMs, several recent works (Qi et al., 2024; Liu et al., 2024a; Zeng et al., 2024; Shalabi et al., 2024) have adopted a different approach by fine-tuning a pretrained MLLM on misinformation datasets, which achieves better performance. However, these methods often carry biases specific to their training data, which are not robust to new, more sophisticated misinformation emerging on social platforms. Therefore, several studies have explored more powerful closed-source MLLMs and have achieved better results. However, these models are either claimed evidence-free (Geng et al., 2024), or evaluated on less updated or real-world datasets (Khaliq et al., 2024; Xuan et al., 2024; Liu et al., 2024b; Jin et al., 2024), raising concerns about their effectiveness when deployed on evolving social media.

# 3 OUR XFACTA DATASET

**Multimodal Misinformation Detection** refers to assessing the authenticity of a news post that includes both supporting images and text. Formally, given a set of supporting images $I = \{I_1, \ldots, I_n\}$ and a text claim $T$, this task is to determine whether the post $\mathcal{P} = (I, T)$ is real or fake.

The supporting images $I$ can make the text claim $T$ seem more believable, even if they are unrelated or misleading, which makes detection much harder than in the unimodal setting. Therefore, most methods incorporate retrieved evidence $\mathcal{E} = (E_i, E_t)$ into their detection pipeline, where $E_i$ and $E_t$ are image-type and text-type evidence, respectively.

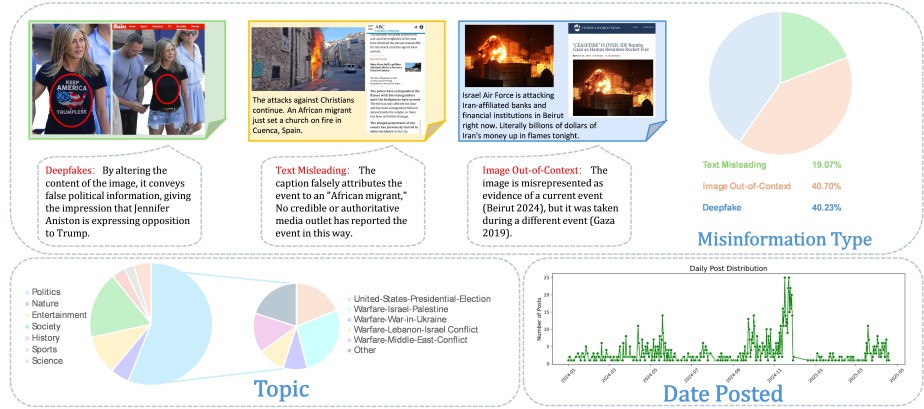

Figure 2: Examples and distribution of misinformation types, topics, and posting dates in XFACTA.

## 3.1 DATA SOURCE & COLLECTION

Our dataset is sourced from X/Twitter. The real news posts are collected from authoritative news organizations including CNN, Fox News, The Guardian, and BBC. The fake news posts are curated from content flagged as false by BBC-certified journalists and X Community Notes.

We first collect fake news posts, as they are rarer and require careful identification, after which we gather about five times more real posts to ensure a diverse selection. This allows us to sample a subset of real posts that matches the fake posts in both quantity and distribution, reducing potential evaluation bias. We guarantee the distribution alignment in two aspects: (1) `Topic-aligned selection`, where we label the topic for both real and fake posts. We then ensure that the number of real and false posts per topic is the same, which helps reduce semantic differences by keeping the content semantically aligned. A detailed description of the topic of posts will be provided in Section 3.2. (2) `Image similarity selection`: the previous step focuses more on aligning the text claims $T$, here we address the alignment of images $I$. We use SigLip (Zhai et al., 2023) to extract image features and apply the Optimal Transport algorithm (Genevay et al., 2016) to select real posts whose image feature distribution most closely matches that of the fake posts. This alignment helps minimize the bias caused by visual differences, ensuring the evaluation accurately reflects the model's true capability in detecting misinformation from both textual and visual perspectives.

In addition, to ensure the reliability of news post labels, beyond the post content $\mathcal{P}$, each is provided with its metadata, including post URL, author id, date, topic, etc. For fake posts, we also collect flagging posts that give reasons and evidence for labeling as fake, while based on flagging posts, we also annotate the misinformation types, with more details provided in Section 3.2. We manually review each entry, and only those with clear evidence of misinformation are included in the dataset.

## 3.2 DATA STATISTICS & ANALYSIS

Our XFACTA dataset contains a total of 2400 data points, including 1200 real posts and 1200 fake posts. For the convenience of model development, we randomly selected 120 real and 120 fake posts as the `Dev` set, while the remaining 2160 posts was used as the `Test` set. As shown in the bottom right corner of Fig. 2, all data are collected from January 2024 to April 2025, with a majority of data collected after September 2024. This contemporary nature ensures the dataset reflects emerging patterns and evolving characteristics of both real and fake news.

To better understand the dataset, we annotate each news post based on its topic and the types of misinformation in fake news. For topic classification, each post $\mathcal{P}$ is categorized into one of the following: *politics, society, entertainment, science, history, nature*, and *sports*, as shown in the bottom left corner of Fig. 2. Notably, political and conflict-related misinformation dominates but is also accompanied by other domains, which aligns closely with current global trends.

For fake posts, as illustrated at the top of Fig. 2, we assign one or more labels from three predefined misinformation types, based solely on explicit evidence provided in the collected flagging posts. We do not assign labels based on inference or assumptions beyond the provided evidence. The three error types are defined as follows:

- **Deepfakes**: The image is generated or digitally manipulated as identified by the flagging post.
- **Image Out-of-Context (OOC)**: The image is authentic but, according to the flagging post, originates from a different event than the one described in the accompanying text. This does not indicate whether the text is true or false.
- **Text Misleading**: The textual content conveys a claim that has been explicitly identified as false by the flagging post. This does not indicate whether the image is authentic or relevant.

By annotating each fake post with these finer-grained misinformation labels, we achieve a more nuanced understanding of the characteristics of multimodal misinformation, and enable a more detailed analysis of a misinformation detector's performance across different misinformation types.

## 4 How to Build a Good MLLM-based Misinformation Detector?

In this section, we explore different design strategies for MLLM-based misinformation detection on the XFACTA dataset. We mainly investigate two questions: (1) How different types of evidence contribute to misinformation detection, and how we can better leverage them; (2) How different LLM reasoning approaches affect the model's prediction.

### 4.1 Analysis of Evidence Retrieval

#### 4.1.1 Experiment Setup

For a given post $\mathcal{P}$ to be verified, we assume the retrieved evidence can assist the detection model in two main aspects: (1) verifying the authenticity of the event described in the post, and (2) verifying whether the accompanying image is used in an out-of-context manner. Based on these assumptions, we introduce eight evidence retrieval strategies designed to support these goals:

- ① **Unimodal Evidence**: Using the post text $T$ to retrieve textual evidence $E_t$ to support Aspect (1). It mimics how humans verify news by searching for relevant information online.
- ②-③ **Cross-modal Evidence**: Using the post text $T$ and image $I$ separately to retrieve image-type evidence $E_i$ (strategy ②) and text-type evidence $E_t$ (stragety ③), following the cross-modal retrieval approach in Abdelnabi et al. (2022) to support Aspect (2).
- ④-⑤ **LLM Querying**: Using an LLM to generate questions about uncertain or suspicious details in the post, then forming search queries to retrieve image-type evidence $E_i$ (strategy ④) and text-type evidence $E_t$ (strategy ⑤). This simulates how humans investigate unclear claims by asking targeted questions.
- ⑥-⑧ **DuckDuckGo Variants**: To explore how different search engines influence retrieval results, we replace the search engine used strategies ⑥ and ⑦ with DuckDuckGo in strategies ① and ②, respectively, We also use DuckDuckGo's "search news" for news evidence $E_{news}$ (strategy ⑧), investigating whether it can retrieve more authoritative evidence.

Additionally, we believe that post-processing can help clean the evidence to reduce its noise. Here, we propose two methods for evidence post-processing inspired by Xuan et al. (2024):

- **Domain Filter**: Filtering out evidence from untrustworthy domains.[1]
- **Evidence Extraction**: Using an MLLM (GPT-4o in our paper) to select parts of the evidence that are highly relevant to the news post and remove irrelevant parts.

To evaluate the impact of each evidence type, we first run the model without evidence, relying only on an MLLM's internal knowledge. Then, we add each of the eight evidence types separately and compare the results against the no-evidence baseline and with each other. We use Chain-of-Thought

---

[1]Evidence from domains used in dataset curation is excluded by default to avoid leakage.

Table 2: Comparison of MLLM Performance with varying evidence retrieval approaches.

| Evidence Type | GPT-4o | | | Gemini-2.0-flash | | | Qwen-vl-7b | | |
|---|---|---|---|---|---|---|---|---|---|
| | Acc. | R. Acc. | F. Acc. | Acc. | R. Acc. | F. Acc. | Acc. | R. Acc. | F. Acc. |
| no evidence | 70.8 | 50.8 | 90.8 | 71.7 | 78.3 | 65 | 60.8 | 76.7 | 44.4 |
| **Google Search** | | | | | | | | | |
| ① $T \rightarrow E_t$ | 87.1 | 97.5 | 76.7 | 81.3 | 98.3 | 64.2 | 59.7 | 82.7 | 38.1 |
| ② $T \rightarrow E_i$ | 81.7 | 75.8 | 87.5 | 77.9 | 90.8 | 65 | 62.2 | 84.2 | 40.2 |
| ③ $I \rightarrow E_t$ | 77.9 | 70 | 85.8 | 78.8 | 83.3 | 74.1 | 55.7 | 71.7 | 39.3 |
| ④ Query $\rightarrow E_i$ | 69.2 | 51.7 | 86.7 | 71.9 | 76.5 | 67.2 | 55.8 | 91.2 | 20 |
| ⑤ Query $\rightarrow E_t$ | 77.5 | 80 | 75 | 77.7 | 83.9 | 71.7 | 56.1 | 63.9 | 48.1 |
| **DuckDuckGo Search** | | | | | | | | | |
| ⑥ $T \rightarrow E_t$ | 84.2 | 94.2 | 74.2 | 79.2 | 97.5 | 60.8 | 64 | 91 | 37 |
| ⑦ $T \rightarrow E_i$ | 76.3 | 64.2 | 88.3 | 76.7 | 87.5 | 65.8 | 53 | 79 | 26.5 |
| ⑧ $T \rightarrow E_{news}$ | 84.2 | 80 | 88.3 | 75.3 | 92.4 | 58.3 | 68.4 | 84.4 | 52.9 |

Table 3: Comparison of different evidence post-processing methods with GPT-4o.

| Method | Acc. | R. Acc. | F. Acc. |
|---|---|---|---|
| $T \rightarrow E_t$ | 87.1 | 97.5 | 76.5 |
| Domain Filter | 88.3 | 98.3 | 78.3 |
| Evidence Extraction | 87.5 | 95.8 | 79.2 |
| $T \rightarrow E_i$ | 81.7 | 75.8 | 87.5 |
| Domain Filter | 83.8 | 80.8 | 86.7 |
| Evidence Extraction | 82 | 77 | 87 |
| $I \rightarrow E_t$ | 77.9 | 70 | 85.8 |
| Domain Filter | 79.6 | 72.5 | 86.7 |
| Evidence Extraction | 81.3 | 75 | 87.5 |

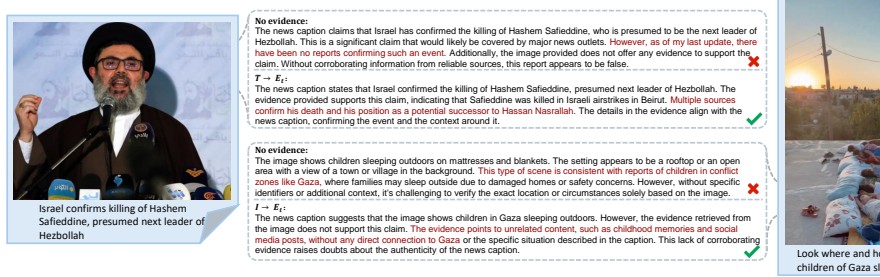

Figure 3: Examples of how $T \rightarrow E_t$ and $I \rightarrow E_t$ correct the no-evidence detection error.

(CoT) (Wei et al., 2022) prompting to obtain interpretable reasoning outputs instead of simple binary decisions. Experiments are conducted on the Dev set using three models with different scales: GPT-4o (Yang et al., 2023), Gemini-2.0-Flash (Team et al., 2023), and Qwen-VL-7B (Wang et al., 2024), to reduce model-specific bias. For post-processing, we test the performance with and without each strategy on GPT-4o. We report three metrics: overall accuracy (Acc.), accuracy on real posts (R. Acc.), and accuracy on fake posts (F. Acc.). The model also outputs a confidence score (0–100), and we report average confidence (Avg. Conf.) in certain tables to reflect prediction certainty.

### 4.1.2 RESULTS AND ANALYSIS

Table 2 and 3 present the performance of different evidence retrieval and post-processing strategies, respectively. Table 4 presents a more detailed comparison across misinformation types for fake posts. We summarize several key observations as follows.

**1. All types of evidence consistently improves accuracy over the no-evidence baseline.** Without evidence, models exhibit notable differences in their behavior: GPT is more conservative, whereas Gemini and Qwen are more inclined to label posts as real. With evidence, their classifications become more balanced, showing the importance of external evidence in misinformation detection.

**2. $T \rightarrow E_t$ (strategy ①) substantially boosts performance, especially for real posts.** This is expected—even for humans, real news is more likely to be supported by online evidence and thus easier to verify. See left of Fig. 3 for example. However, accuracy for fake posts does not notably improve, and even declines slightly for GPT-4o. We attribute this to OOC misinformation, where $T \rightarrow E_t$ provides no information about the image $I$, and the strong support for $T$ in $E_t$ misleads the model to flip its originally correct prediction of fake.

**3. $I \rightarrow E_t$ (strategy ③) is more effective than $T \rightarrow E_i$ (strategy ②) for out-of-context misinformation.** Although $T \rightarrow E_i$ shows higher overall accuracy in Table 2, manual inspection reveals that $I \rightarrow E_t$ better detects out-of-context cases. This is because $I \rightarrow E_t$ retrieves webpages directly containing the query image and extracts highly relevant text, while $T \rightarrow E_i$ conducts a fuzzy search based on the caption and often retrieves loosely related images. In addition, textual evidence is also more informative in these cases, since image-based comparisons are often limited to coarse features

Table 4: Comparison of each evidence retrieval strategies across misinformation types with GPT-4o.

| Evidence Type | Deepfakes | | Image OOC | | Text Misleading | |
|---|---|---|---|---|---|---|
| | Acc. | Avg. Conf. | Acc. | Avg. Conf. | Acc. | Avg. Conf. |
| no evidence | 89.7 | 87.4 | 93.8 | 77 | 91.1 | 82.6 |
| Google Search | | | | | | |
| ① $T \to E_t$ | 79.3 | 87.8 | 77.1 | 84 | 80.4 | 87 |
| ② $T \to E_i$ | 93.1 | 85.9 | 85.4 | 81 | 87.5 | 83 |
| ③ $I \to E_t$ | 100 | 88.5 | 83.3 | 85.2 | 80.4 | 88.2 |
| ④ Query $\to E_i$ | 89.7 | 88.8 | 79.2 | 82.9 | 89.3 | 85.4 |
| ⑤ Query $\to E_t$ | 86.2 | 91.2 | 60.4 | 88.7 | 82.1 | 89.6 |
| DuckDuckGo Search | | | | | | |
| ⑥ $T \to E_t$ | 79.3 | 90 | 64.6 | 85.5 | 82.1 | 86.1 |
| ⑦ $T \to E_i$ | 89.7 | 85.9 | 83.3 | 79.6 | 91.1 | 80.9 |
| ⑧ $T \to E_{news}$ | 93.1 | 84.8 | 81.3 | 80.6 | 92.9 | 80.8 |

Table 5: Comparison of MLLM Performance with various reasoning methods on the Dev set.

| Reasoning Method | GPT-4o | | | Gemini-2.0-flash | | | Qwen-vl-7b | | |
|---|---|---|---|---|---|---|---|---|---|
| | Acc. | R. Acc. | F. Acc. | Acc. | R. Acc. | F. Acc. | Acc. | R. Acc. | F. Acc. |
| Chain of Thought | 88.3 | 98.3 | 78.3 | 83.8 | 98.3 | 69.2 | 54.8 | 84.2 | 24.1 |
| Prompt Ensembles | 90 | 100 | 80 | 85.4 | 98.3 | 72.5 | 67.1 | 90 | 44 |
| Self Consistency | 88.3 | 97.5 | 79.2 | 86.7 | 98.3 | 75 | 61 | 64 | 58.2 |
| Multi-step Reasoning | 91.3 | 91.7 | 90.8 | 81.3 | 90 | 72.5 | 62.1 | 78.4 | 45.9 |

like general scenes or people. These superficial similarities are usually preserved in out-of-context misinformation, making it hard to detect manipulation through image-type evidence. Point 5 further analyzes strategy ③ on fake posts.

**4. LLM-generated queries (strategies ④ and ⑤) are less effective than direct caption searches.** In most cases, an LLM is not able to generate highly targeted queries; most of them are simply paraphrases of the original caption. Searching with such paraphrased versions is thus less accurate than directly using the caption $T$ itself to retrieve evidence. In certain cases of fake posts, if the questions or doubts raised by the LLM fail to target the actual reason why the post is fake, the retrieved evidence can even lead the model to confidently make an incorrect judgment, as further analyzed in the next point.

**5. Across different misinformation types, $I \to E_t$ (strategy ③) provides consistently informative evidence especially for identifying fake posts.** Unlike earlier analyses based on the overall accuracy, analyzing fine-grained misinformation types for fake posts demands careful consideration beyond accuracy alone. GPT-4o tends to conservatively classify ambiguous posts as fake without additional evidence, and this can inflate the accuracy of fake posts. Hence, average confidence scores become essential because they indicate whether the retrieved evidence provides clear and informative knowledge that truly helps the model's judgment. As shown in Table 4, strategy ③ not only achieves high accuracy but also consistently maintains high confidence across Deepfakes and Image OOC categories. Although Query $\to E_t$ (strategy ⑤) shows slightly superior combined performance in the Text Misleading category, it causes the model to make highly confident but incorrect predictions in Image OOC cases, significantly reducing its overall utility. Therefore, $I \to E_t$ remains the optimal evidence retrieval strategy across various misinformation types.

**6. DuckDuckGo provides lower-quality evidence than Google (strategis ⑥-⑧).** Experiments indicate that evidence retrieved using DuckDuckGo consistently yields lower performance compared to Google Search. Additionally, $T \to E_{news}$ did not produce the expected improvements.

**7. Domain Filter can mitigate evidence noise.** As shown in Table 3, domain filter can improve accuracy in general, suggesting that evidence from low-credibility websites is indeed noisy and potentially misleading.

**8. LLM-Based Evidence Extraction can mitigate evidence noise.** We inspected the extraction results and found that the LLM can successfully retain the key information needed to detect misinformation and filter out some irrelevant evidence, which leads to improved detection accuracy as shown in Table 3, especially for $I \to E_t$. However, it is important to note that evidence extraction introduces a huge token overhead.

Table 6: Comparison of detection performance across different LLMs on the `Test` set.

| Model | Scale | Acc. | R. Acc. | F. Acc. |
|---|---|---|---|---|
| **GPT** | GPT-4o-mini | 83 | 84.6 | 81.3 |
| | GPT-4o | 88.6 | 87.6 | 89.6 |
| **Gemini** | Gemini-2.0-lite | 76.2 | 77.2 | 75.2 |
| | Gemini-2.0-flash | 78.9 | 83.6 | 74.2 |
| **Qwen** | Qwen-vl-7b | 65 | 80.9 | 48.5 |
| | Qweb-vl-72b | 81 | 82.3 | 79.6 |

Table 7: Comparison of different misinformation detection methods on the `Test` set. *: 284 samples are excluded due to the ambiguous output.

| Methods | Training Set / LLM | Acc. | R. Acc. | F. Acc. |
|---|---|---|---|---|
| **SENs** | NewsCLIPpings | 52.6 | 58.4 | 46.7 |
| **Mocheg** | Mocheg | 51.7 | 57.6 | 45.5 |
| **HAMMER** | DGM[4] | 57.2 | 78.2 | 36.1 |
| **Sniffer** | NewsCLIPpings | 56.1 | 71 | 41.1 |
| **MMFakeBench**[*] | GPT-4o | 68.2 | 61.5 | 75.6 |
| **LEMMA** | GPT-4o | 77.3 | 63.9 | 90.8 |
| **Ours** | GPT-4o | 88.8 | 87.2 | 90.4 |

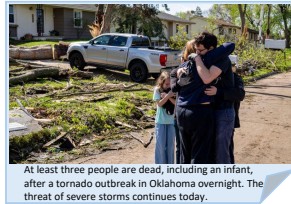

At least three people are dead, including an infant, after a tornado outbreak in Oklahoma overnight. The threat of severe storms continues today.

**CoT Reasoning:**
The news caption states that at least three people, including an infant, died in a tornado outbreak in Oklahoma. The textual evidence from both image and caption searches confirms that multiple tornadoes struck Oklahoma, resulting in at least four deaths, including an infant. The reports also mention severe weather conditions continuing, aligning with the news caption. The image shows a scene consistent with tornado aftermath, supporting the context of the news. Therefore, the news report is consistent with the available evidence. ✓

**Multi-Step Reasoning:**
[Step 1: Caption Authenticity]
Predicted: True
Reasoning: ......
[Step 2: Image Misuse Check]
Predicted misuse: True
Reasoning: The news caption describes a tornado outbreak in Oklahoma resulting in fatalities, including an infant. The image shows people embracing amidst debris, which aligns with the aftermath of a tornado. However, the text evidence indicates that the image is from a tornado event in Nebraska and Iowa, not Oklahoma. Therefore, the image does not correspond to the specific event described in the news caption.
[Final Decision]
Predicted Label: False
Reasoning: The news caption is accurate regarding the tornado outbreak in Oklahoma, with confirmed fatalities, including an infant. However, the image is misused as it depicts a different tornado event in Nebraska and Iowa. This misuse could lead readers to associate the image with the Oklahoma event, potentially causing misunderstanding about the location and context of the damage shown. Therefore, the overall news report is considered inauthentic due to the misleading use of the image. ✗

Figure 4: Multi-step reasoning can detect image "misuse" in the news post from CNN. We believe this "overly strict" behavior is actually beneficial for reliable misinformation detection.

## 4.2 ANALYSIS OF REASONING

### 4.2.1 EXPERIMENT SETUP

We use $T \to E_t$ (strategy ①) and $I \to E_t$ (strategy ③) in the reasoning stage, as they can complement each other well. We also apply domain filter to reduce evidence noise, but skip evidence extraction to better assess the reasoning ability on noisy evidence pieces. We test four reasoning strategies, including **CoT** (Wei et al., 2022), **Prompt Ensembles** (Geng et al., 2024), **Self Consistency** (Wang et al., 2022), and **Multi-step Reasoning**. See the appendix for additional details.

### 4.2.2 RESULTS AND ANALYSIS

We summarize several key observations below according to the results reported in Table 5.

**1. The stronger the MLLM, the less it is affected by different reasoning methods.** Stronger models like GPT-4o usually have good reasoning ability by default and have similar accuracy across different reasoning techniques.

**2. Different model architectures show different preferences for different reasoning methods.** Therefore, in practice, deploying an MLLM-based misinformation detector should involve testing various reasoning methods, especially for smaller MLLMs, to achieve better performance.

**3. For GPT-4o, multi-step reasoning has the overall best balanced accuracy.** For the best performing model GPT-4o, by manually inspecting the reasoning paths across various strategies, we find that multi-step reasoning consistently provides the clearest and most structured reasoning. Particularly, its accuracy in detecting fake posts is superior to other methods. However, its accuracy on real posts is not that good. Interestingly, we found that some real posts from reputable news sources may use images from unrelated events (which we do not consider as misinformation because there is no intention to mislead). Multi-step reasoning can identify and flag these cases as fake due to image mismatch. We believe this "overly strict" behavior is actually beneficial for reliable misinformation detection. An example can be found in Fig. 4.

## 5 FURTHER EVALUATIONS ON XFACTA

**Comparison of Different MLLMs.** We evaluate the performance of various MLLMs on our `Test` set. Specifically, we analyze performance differences across closed-source models (GPT and Gemini), as well as open-source models (Qwen), both across different model scales. We use the evidence

types following Section 4.2 and use the multi-step reasoning strategy. Results are shown in Table 6. For the same model architecture, larger models always achieve higher accuracy.

**Comparison of Existing Multimodal Misinformation Detection Methods.** We perform a horizontal comparison with existing multimodal misinformation approaches using our Test set, including models trained from scratch: SENs (Yuan et al., 2023), Mocheg (Yao et al., 2023), and HAMMER (Shao et al., 2023); methods fine-tuning MLLMs: Sniffer (Qi et al., 2024), and zero-shot methods using closed-source MLLMs: MMFakeBench (Liu et al., 2024b), LEMMA (Xuan et al., 2024). Results in Table 7 show that specialist methods that trained on a specific dataset suffer from severe generalization issues. In contrast, models that use GPT-4o demonstrate relatively good performance. Based the systematic analysis of evidence retrieval and reasoning strategies, our method outperforms these models, establishing SOTA accuracy on XFACTA.

**Comparison of Similar Datasets.** We examine the dependency on evidence across similar datasets. To achieve this, we select a subset from (Papadopoulos et al., 2023), Snopes+Reuters Zlatkova et al. (2019), and NewsCLIPpings (Luo et al., 2021) dataset, respectively. We use the evidence types following Section 4.2 and apply CoT reasoning with GPT-4o. We report the results in Table 8 in the Appendix. Notably, for our XFACTA dataset, GPT-4o does not work well without any evidence, confirming the need for a contemporary, real-world benchmark.

**Detector Effectiveness on More Recent and Out-of-Distribution Data.** We evaluate whether the misinformation detector trained on the original XFACTA dataset remains effective when applied to more recent and out-of-distribution social media content. First, we choose Snopes as the testbed since, compared to X, it is out-of-distribution. Moreover, the website provides real/fake annotations which are provided by professional journalists, thus can serve as a reference for evaluating the performance of the detector. We collect 1,200 fact-checked news items (600 real and 600 fake) from Snopes between July 2024 and July 2025, which are more recent than original XFACT dataset. Our resulting detector achieves an overall accuracy of 89.2% on this dataset (85.5% on true and 93.0% on false), showing that the model works well on both newest and out-of-distribution data.

# 6    CLOSING THE LOOP: DETECTOR-ASSISTED DATASET EXPANSION

In this section, we demonstrate how our detector, which has been validated on the original XFACTA dataset, can be effectively used to support dataset expansion. Previous experiments show that the detector maintains stable performance on both more recent and out-of-distribution data, suggesting that it generalizes well to continuously emerging, previously unseen content. Therefore, it can be integrated into the dataset collection pipeline to assist human reviewers in verifying misinformation, enabling a detection-in-the-loop framework that accelerates and scales up the data curation process.

To verify this idea, we conduct a case study as a proof of concept. This time, we do not rely on journalist-flagged posts or posts from official news accounts for real/fake references. Instead, we select several accounts that regularly post about trending or controversial topics and that have a sizable follower base. We crawl and identify 500 posts between June 2025 and July 2025 from these accounts. Among them, 265 posts are identified by our detector as fake and 235 as real. For each prediction, the detector also generates an explanation to support its decision. The explanations can assist human reviewers in verifying the predictions more efficiently and deciding whether to incorporate the posts into XFACTA. An example can be found in Appendix. The additional dataset has been released alongside the main XFACTA dataset as part of our public release.

# 7    CONCLUSION

In this paper, we introduced XFACTA, a contemporary, real-world dataset for multimodal misinformation detection. Using this dataset, we analyze how to build an effective MLLM-based misinformation detector from two perspectives: evidence retrieval and reasoning. Our experiments offer practical insights into developing robust detection systems. Furthermore, we implement a semi-automatic detection-in-the-loop cycle to continuously update XFACTA with newly flagged content. We also benchmark SOTA MLLMs and existing detection methods in a more realistic setting using our dataset. We believe that XFACTA and our findings will foster future research in multimodal misinformation detection.

ETHICS STATEMENT

Our research adheres to the guidelines set forth by the Twitter Developer Terms[2]. We ensure that our data collection and use comply with these terms, including the appropriate use of the Twitter API. While we plan to release our dataset for research purposes, we will do so in a manner that adheres to all applicable rules and guidelines.

Our study focuses on detecting multimodal misinformation, a significant issue in the digital age. By identifying and mitigating the spread of misinformation, our work contributes positively to the integrity of information on the web. Since our dataset consists of internet fake news posts, some posts may contain offensive content. However, the positive contributions of our research in reducing misinformation far outweigh the potential negatives.

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

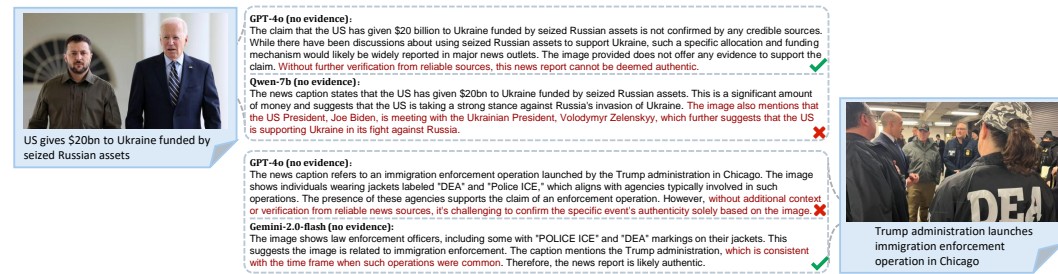

Figure 5: Comparisons of different MLLMs' performance without evidence. Different MLLMs exhibit distinct prediction tendencies in the absence of supporting evidence.

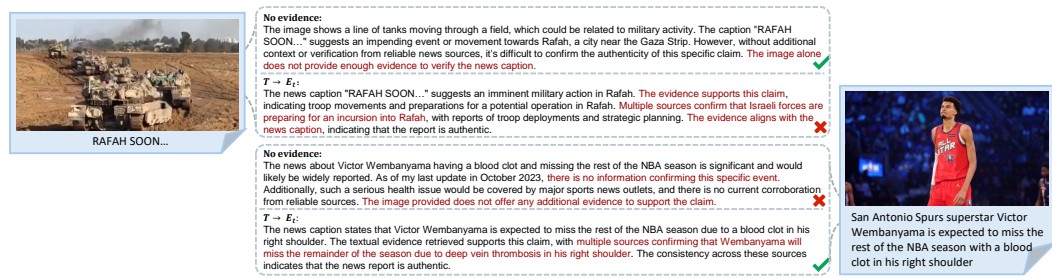

Figure 6: Effectiveness of $T \to E_t$ strategy on real and fake posts. $T \to E_t$ is good at finding evidence for real posts. However, it may also leads the model to flip its originally correct prediction, particularly for image OOC misinformations.

## A    MORE ANALYSIS OF EVIDENCE RETRIEVAL

**MLLMs exhibit notable differences in their behavior without evidence.** As shown in Fig. 5, GPT shows a conservative tendency in multimodal misinformation detection. Whether the post is fake (as shown in the upper figure) or real (as in the lower figure), GPT tends to classify it as fake when no supporting evidence is available. In contrast, Gemini and Qwen exhibit the opposite behavior: they are more likely to classify the news as real if no clear inconsistency is observed between the image and the caption. This further highlights that relying solely on the model's internal knowledge, without external evidence, is unreliable for misinformation detection.

$T \to E_t$ **(strategy ①) substantially boosts performance, especially for real posts.** Two additional examples are shown in the bottom of Fig. 6 and the top of Fig. 7. We then discuss why the accuracy for fake posts does not notably improve, and even declines slightly for GPT-4o. We attribute this to image OOC misinformation, where $T \to E_t$ provides no information about the image $I$, and the strong support for $T$ in $E_t$ misleads the model to flip its originally correct prediction of fake. As illustrative examples, two cases are shown in Fig. 8 and at the top of Fig. 6. Without evidence, the model gives a cautious and right answer, while with $T \to E_t$ supporting the post claim $T$, it becomes more confident but makes a wrong prediction. Therefore, evidence that directly targets image OOC misinformation serves as an important complement to this evidence such as the example shown on the right side of Fig. 3.

$I \to E_t$ **(strategy ③) is more effective than** $T \to E_i$ **(strategy ②) for out-of-context misinformation.** Although $T \to E_i$ shows higher overall accuracy in Table 2, manual inspection reveals that $I \to E_t$ better detects out-of-context cases. This is because $I \to E_t$ retrieves webpages directly containing the query image and extracts highly relevant text, while $T \to E_i$ conducts a fuzzy search based on the caption and often retrieves loosely related images. Textual evidence is also more informative in these cases, since image-based comparisons are often limited to coarse features like general scenes or people. These surface-level similarities are usually preserved in out-of-context misinformation, making it hard to detect manipulation through image evidence alone. As shown in Fig. 8, $I \to E_t$ correctly traces the image to a Facebook user and identifies it as out-of-context mis-

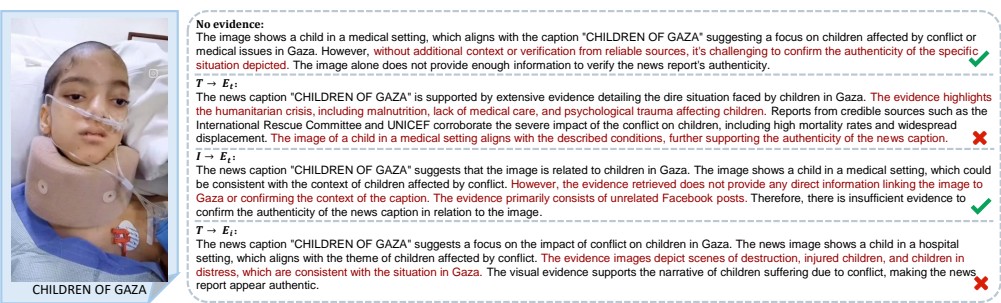

Figure 7: Effectiveness of different evidence types on real posts. $T \rightarrow E_t$ can effectively retrieve relevant evidence for real posts, but cross-modal evidence is less useful in this case.

Figure 8: Effectiveness of different evidence types on out-of-context misinformation posts.

information. In contrast, $T \rightarrow E_i$ provides only rough background comparison, failing to precisely detect the misinformation.

**DuckDuckGo provides lower-quality evidence than Google.** We present one example here in Fig. 10, where DuckDuckGo News Search (strategy ⑧) failed to retrieve any relevant evidence, while Google Search (strategy ①) accurately returned the "direct match" news events that helped the model make the correct inference.

**Domain Filter can mitigate evidence noise.** Domain filter can improve accuracy by filtering out misleading evidence from low-credibility websites. Here we give an example. As shown in Fig. 9, without the domain filter, there is relevant content in the evidence that supports the claim made in the caption. Therefore, even though the news post caption itself mentions that this might be an "unconfirmed rumor", the model still classifies it as true based on the supporting evidence. However, the domain filter excludes this incorrect evidence, making it easy for the model to make the correct judgment. This shows that incorrect evidence can still have a significant negative impact on detection, even in cases where the correct classification should be straightforward.

**LLM-Based Evidence Extraction can mitigate evidence noise.** We present an example in Fig. 11. The extracted evidence is more concise, making it easier for the model to perform reasoning.

## B    MORE ANALYSIS OF REASONING

**Details of reasoning methods:**

- **Chain of Thought**: Model outputs an additional rationale in addition to the binary label.
- **Prompt Ensembles**: Inspired by Geng et al. (2024), we use a variety of prompts to generate multiple responses, then ask the model to aggregate the responses to get a more robust result.
- **Self Consistency**: Perform multiple rounds of inference and use majority vote to obtain the final result.
- **Multi-step Reasoning**: The model may become confused when multiple sources of evidence are provided. Therefore, we invoke the LLM separately for each type of evidence and then summarize all intermediate reasoning processes to produce a final aggregated answer.

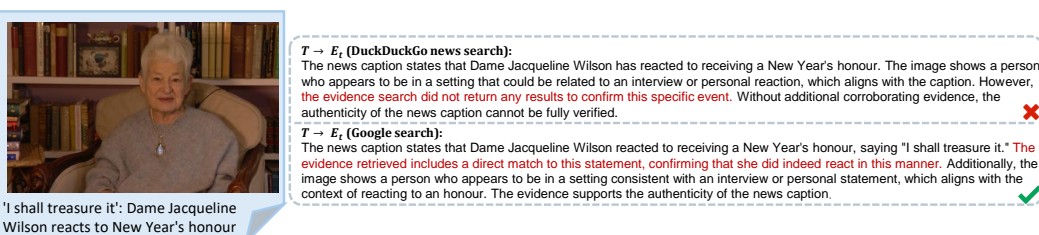

Figure 9: Effectiveness of evidence domain filter.

Figure 10: Comparison between Google Search and DuckDuckGo News Search.

**Comparison between CoT and multi-step reasoning.** Figure 12 shows the different reasoning paths between CoT reasoning and multi-step reasoning. In this example, multi-step reasoning accurately identifies that the image originates from another event by analyzing $I \rightarrow E_t$, leading to the correct classification as image OOC misinformation. However, the CoT reasoning fails to fully utilize each piece of evidence, leading it to overlook the $I \rightarrow E_t$ evidence and resulting in an incorrect inference.

**Comparison of reasoning performance across different model sizes.** Figure 13 presents the reasoning paths of GPT-4o and GPT-4o-mini. GPT-4o has stronger reasoning capabilities than GPT-4o-mini, which allows it to more precisely recognize the phrase "initially entered a not-guilty plea" in the evidence and therefore make the correct judgement.

## C COMPARISON OF DIFFERENT MISINFORMATION DETECTION DATASETS

Results are shown in Table 8. GPT-4o achieves an accuracy of 0.8 or even 0.9 without using any evidence on other datasets, indicating that it can perform misinformation detection effectively through memorization alone. Moreover, we observe that the improvement brought by evidence is most significant on our dataset. Therefore, our dataset is more suitable for evaluating retrieval-based misinformation detectors and has less evaluation bias compared to real-world misinformation scenarios.

## D EXAMPLE OF DETECTOR-ASSISTED DATASET EXPANSION

Fig. 14 shows an example of misinformation detection on the newest posts from X using our detector, including the supporting explanations for each prediction to assist human reviewers in verifying the results more efficiently.

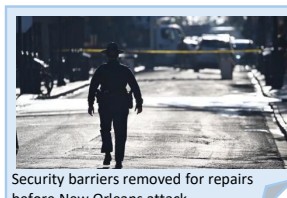

**$I \rightarrow E_t$ (w/o evidence extraction):**
The news caption states that security barriers were removed for repairs before an attack in New Orleans. The evidence provided discusses a truck attack in New Orleans, mentioning that the attacker acted alone and was inspired by ISIS. However, there is no specific mention of security barriers being removed for repairs in the evidence. The image shows a street with road work signs, but this does not directly confirm the removal of security barriers for repairs. The lack of direct evidence supporting the specific claim about the removal of barriers makes the news caption questionable. ✗

**$I \rightarrow E_t$ (w/ evidence extraction):**
The news caption states that security barriers were removed for repairs before an attack in New Orleans. The textual evidence supports the occurrence of an attack on New Year's Day 2025, where a lone attacker drove a truck into a crowd on Bourbon Street, resulting in multiple casualties. The evidence also mentions discussions about enhancing security measures, such as installing permanent barriers, which aligns with the news caption's implication of security concerns. The image shows a street with road work signs and barriers, consistent with the context of repairs and security measures. Therefore, the news caption is consistent with the textual evidence and the image. ✓

Security barriers removed for repairs before New Orleans attack

Figure 11: Effectiveness of evidence extraction.

| Dataset | Evidence | Acc. | R. Acc. | F. Acc. |
|---|---|---|---|---|
| **VERITR** | ✗ | 80.1 | 78 | 82 |
| | ✓ | 91.9 | 88.5 | 95.2 |
| **Snopes+Reuters** | ✗ | 91.9 | 93.5 | 90.3 |
| | ✓ | 96.7 | 94.6 | 99 |
| **Newsclipping** | ✗ | 80.7 | 88.2 | 73.2 |
| | ✓ | 89.5 | 91.2 | 87.7 |
| **XFACTA(Ours)** | ✗ | 70.5 | 51 | 90 |
| | ✓ | 89.5 | 98 | 81 |

Table 8: Comparison of GPT-4o's performance on different datasets.

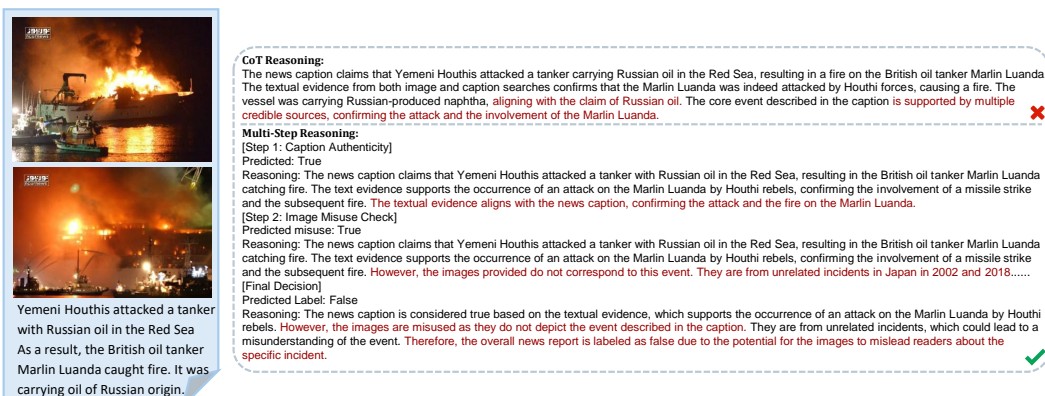

**CoT Reasoning:**
The news caption claims that Yemeni Houthis attacked a tanker carrying Russian oil in the Red Sea, resulting in a fire on the British oil tanker Marlin Luanda. The textual evidence from both image and caption searches confirms that the Marlin Luanda was indeed attacked by Houthi forces, causing a fire. The vessel was carrying Russian-produced naphtha, aligning with the claim of Russian oil. The core event described in the caption is supported by multiple credible sources, confirming the attack and the involvement of the Marlin Luanda. ✗

**Multi-Step Reasoning:**
[Step 1: Caption Authenticity]
Predicted: True
Reasoning: The news caption claims that Yemeni Houthis attacked a tanker with Russian oil in the Red Sea, resulting in the British oil tanker Marlin Luanda catching fire. The text evidence supports the occurrence of an attack on the Marlin Luanda by Houthi rebels, confirming the involvement of a missile strike and the subsequent fire. The textual evidence aligns with the news caption, confirming the attack and the fire on the Marlin Luanda.
[Step 2: Image Misuse Check]
Predicted misuse: True
Reasoning: The news caption claims that Yemeni Houthis attacked a tanker with Russian oil in the Red Sea, resulting in the British oil tanker Marlin Luanda catching fire. The text evidence supports the occurrence of an attack on the Marlin Luanda by Houthi rebels, confirming the involvement of a missile strike and the subsequent fire. However, the images provided do not correspond to this event. They are from unrelated incidents in Japan in 2002 and 2018......
[Final Decision]
Predicted Label: False
Reasoning: The news caption is considered true based on the textual evidence, which supports the occurrence of an attack on the Marlin Luanda by Houthi rebels. However, the images are misused as they do not depict the event described in the caption. They are from unrelated incidents, which could lead to a misunderstanding of the event. Therefore, the overall news report is labeled as false due to the potential for the images to mislead readers about the specific incident. ✓

Yemeni Houthis attacked a tanker with Russian oil in the Red Sea As a result, the British oil tanker Marlin Luanda caught fire. It was carrying oil of Russian origin.

Figure 12: Comparison between CoT reasoning and Multi-step reasoning.

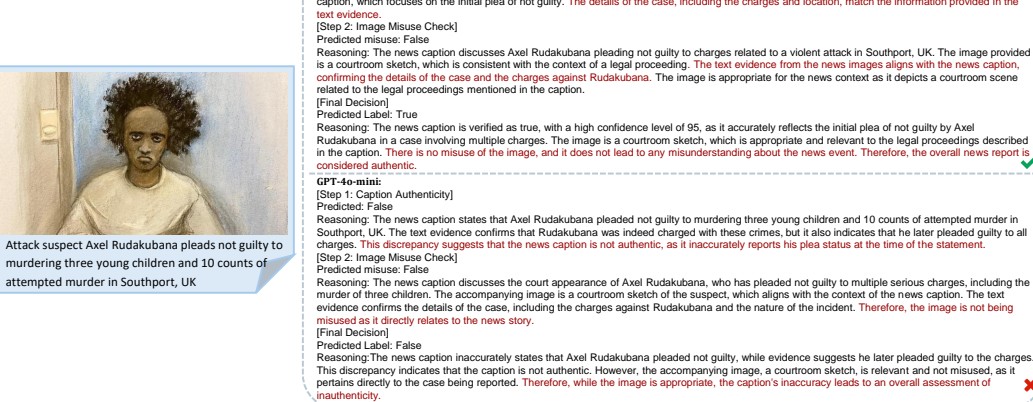

**GPT-4o:**
[Step 1: Caption Authenticity]
Predicted: True
Reasoning: The news caption states that Axel Rudakubana pleaded not guilty to murdering three young children and 10 counts of attempted murder in Southport, UK. The text evidence confirms that Rudakubana initially entered a not-guilty plea before later changing it to guilty. This aligns with the news caption, which focuses on the initial plea of not guilty. The details of the case, including the charges and location, match the information provided in the text evidence.
[Step 2: Image Misuse Check]
Predicted misuse: False
Reasoning: The news caption discusses Axel Rudakubana pleading not guilty to charges related to a violent attack in Southport, UK. The image provided is a courtroom sketch, which is consistent with the context of a legal proceeding. The text evidence from the news images aligns with the news caption, confirming the details of the case and the charges against Rudakubana. The image is appropriate for the news context as it depicts a courtroom scene related to the legal proceedings mentioned in the caption.
[Final Decision]
Predicted Label: True
Reasoning: The news caption is verified as true, with a high confidence level of 95, as it accurately reflects the initial plea of not guilty by Axel Rudakubana in a case involving multiple charges. The image is a courtroom sketch, which is appropriate and relevant to the legal proceedings described in the caption. There is no misuse of the image, and it does not lead to any misunderstanding about the news event. Therefore, the overall news report is considered authentic. ✓

**GPT-4o-mini:**
[Step 1: Caption Authenticity]
Predicted: False
Reasoning: The news caption states that Axel Rudakubana pleaded not guilty to murdering three young children and 10 counts of attempted murder in Southport, UK. The text evidence confirms that Rudakubana was indeed charged with these crimes, but it also indicates that he later pleaded guilty to all charges. This discrepancy suggests that the news caption is not authentic, as it inaccurately reports his plea status at the time of the statement.
[Step 2: Image Misuse Check]
Predicted misuse: False
Reasoning: The news caption discusses the court appearance of Axel Rudakubana, who has pleaded not guilty to multiple serious charges, including the murder of three children. The accompanying image is a courtroom sketch of the suspect, which aligns with the context of the news caption. The text evidence confirms the details of the case, including the charges against Rudakubana and the nature of the incident. Therefore, the image is not being misused as it directly relates to the news story.
[Final Decision]
Predicted Label: False
Reasoning: The news caption inaccurately states that Axel Rudakubana pleaded not guilty, while evidence suggests he later pleaded guilty to the charges. This discrepancy indicates that the caption is not authentic. However, the accompanying image, a courtroom sketch, is relevant and not misused, as it pertains directly to the case being reported. Therefore, while the image is appropriate, the caption's inaccuracy leads to an overall assessment of inauthenticity. ✗

Attack suspect Axel Rudakubana pleads not guilty to murdering three young children and 10 counts of attempted murder in Southport, UK

Figure 13: Comparison between GPT-4o and GPT-4o-mini.

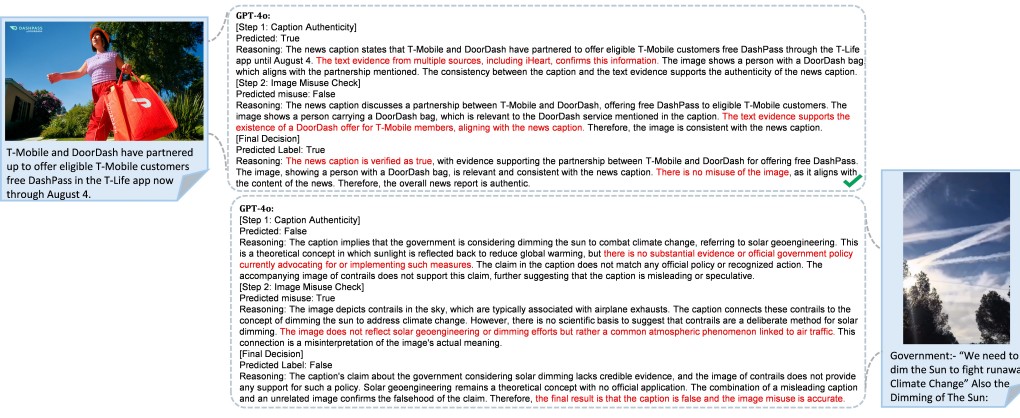

Figure 14: Example of misinformation detection on newest posts from X using the detector's inference

