# OpenReview forum: "XFacta: Contemporary, Real-World Dataset and Evaluation for Multimodal Misinformation Detection with Multimodal LLMs"
_ICLR.cc/2026/Conference — Submitted to ICLR 2026_

### Official Review · Reviewer_WoLQ · 2025-10-27

**Soundness:** 2
**Presentation:** 2
**Contribution:** 2
**Rating:** 4
**Confidence:** 4

**Summary:**

This paper introduces XFACTA, a contemporary, real-world dataset for evaluating multimodal misinformation detection with multimodal large language models (MLLMs). Unlike existing datasets which are outdated, synthetic, or biased. XFACTA collects 2,400 real posts (1,200 real + 1,200 fake) from X/Twitter, spanning January 2024 to April 2025. The authors benchmark closed-source (GPT-4o, Gemini-2.0-Flash) and open-source (Qwen-VL) MLLMs under different evidence retrieval and reasoning strategies, comparing their results with prior detectors such as MMFakeBench, LEMMA, and Sniffer.

**Strengths:**

1. The dataset intentionally focuses on recent events (post-2024), avoiding overlap with pre-training corpora of MLLMs like GPT-4o and Gemini. This makes XFACTA a valuable benchmark for testing real-time misinformation detection, where memorization bias is minimized.
2. The paper systematically investigates multiple dimensions, i.e., evidence retrieval strategies, reasoning paradigms (CoT, multi-step reasoning, etc.), and model architectures, offering insights into where MLLMs struggle (retrieval vs. reasoning bottlenecks).

**Weaknesses:**

1. Unclear labeling process for the three misinformation types. The labeling of Deepfakes, Image Out-of-Context, and Text Misleading relies on “flagging posts by journalists or X Community Notes,” but the paper does not specify who performed the final verification or how inter-annotator reliability was measured.
2. While XFACTA’s realism is commendable, 2,400 samples (1,200 fake) is relatively small compared to datasets. Such a small dataset may not adequately represent the diversity of misinformation strategies on social media.
3. Prompt sensitivity and fair comparison issues. The performance gap in Table 7 between GPT-4o (ours: 88.8%) and other GPT-4o-based systems (e.g., LEMMA 77.3%, MMFakeBench 68.2%) may stem partly from prompt-format differences, not purely methodological superiority. A sensitivity analysis of prompts quality is necessary to ensure that the improvements are reproducible rather than prompt-dependent.
4. Limited depth in experimental validation. Although multiple pretrained models are tested, the study lacks fine-tuning experiments or ablation analyses.
5. Unclear statistics on overlapping or multi-label posts. The paper states that “each fake post may be assigned one or more error-type labels,” but the extent of label overlap is not quantified.

**Questions:**

6. Unclear statistics on overlapping or multi-label posts. The paper states that “each fake post may be assigned one or more error-type labels,” but the extent of label overlap is not quantified.

---

> ### Author Response · Authors · 2025-11-22
>
> We appreciate the effort and time you have devoted to reviewing our paper. Below are our point-by-point responses to your comments.
>
> 1.**Unclear Labeling Procedure for Misinformation Types** (Response to Weakness 1)
>
> Thank you for pointing out this question. To clarify, your comment can refer to the final verification and reliability of (a) whether a post is misinformation, (b) the categorization into the three misinformation types (Deepfake, Image Out-of-Context, Text Misleading). Therefore, we address both components:
>
> （1）The labels in our dataset come from the public judgments made by journalists or X Community Notes on potentially misleading posts. These sources have already conducted professional, transparent, and verifiable fact-checking, so we treat their judgments as the final authoritative labels. Our manual review mainly refers to data-level cleaning, such as deduplication, text and image denoising, confirming the authenticity of journalist accounts, and filtering out fake posts that lack sufficient evidence.
>
> （2）We first defined clear classification criteria for the three misinformation types (Deepfake, Image Out-of-Context, and Text Misleading). Because the journalists who flagged each post as fake provided both reasoning and evidence, we gave this information along with the original post to GPT and prompted it to assign each fact-checked post to one of the three categories according to our rules. GPT was used solely for this three-way categorization and did not make any decisions about the authenticity of the content.
>
>
> For these reasons, our labeling pipeline does not involve multiple annotators independently judging the same content, and thus inter-annotator agreement is not applicable. The reliability of the posts relies on the public fact-checks provided by journalists and Community Notes, while the three error types are given via rule-based automated classification.
>
> 2.**Dataset Size May Be Insufficient for Diversity** (Response to Weakness 2)
>
> We acknowledge that XFacta is smaller than many large-scale automatically collected social media datasets. However, high-quality contemporary misinformation annotated by professional journalists is extremely rare on the open internet. The primary constraint is not data scraping, but the scarcity of reliable, expert-verified signals that meet real-world editorial standards. Because of this, datasets with this level of authenticity and annotation quality naturally remain limited in size. In our case, every data point in XFacta is flagged by trained journalists following established newsroom practices. This yields ground truth that is far more trustworthy than large-scale, automatically collected or heuristically labeled misinformation datasets. In addition, within this size, we deliberately constructed XFacta to balance coverage across major domains (e.g., politics, public health, economics, science, and international affairs), rather than concentrating on a few trending stories or rumor types. This design avoids topic skew and ensures that models are tested on a broad spectrum of contemporary misinformation themes.
>
> For zero-shot evaluation, the stability of the results depends more on annotation precision and diversity of real-world cases than on raw scale. In our experiments, the current size is already sufficient to produce consistent and meaningful comparisons across models. We will add a clarification in the paper to better emphasize this trade-off between dataset scale and annotation authenticity, and why XFacta aims to provide high-precision contemporary misinformation signals rather than maximizing size.

---

> > ### Author Response · Authors · 2025-11-22
> >
> > 3.**Prompt Sensitivity Threatens Fair Comparison** (Response to Weakness 3)
> >
> > We thank the reviewer for pointing out the potential influence of prompt design. To quantify this effect, we conducted a prompt-sensitivity study on GPT-4o using three prompt templates: (1) our original prompt (P_ours), (2) a reimplementation of LEMMA’s prompt (P_LEMMA), and (3) a reimplementation of MMFakeBench’s prompt (P_MMFB). Since our model setup is not fully identical to these methods, during reimplementation, we have to remove prompt that were incompatible with our pipeline (e.g.,  LEMMA requires the model to decide whether it needs external knowledge, but our setup doesn’t include that mechanism, so we removed that part). However, we managed to match their core components as closely as possible, including the basic prompt structure, the definition of misinformation, and the writing style for caption, image, and evidence descriptions. For each prompt template, we evaluated two settings: (a) Direct, where the model directly classifies the sample without any auxiliary evidence, and (b) With evidence, where we apply our evidence extraction pipeline and feed the resulting structured evidence into the classification prompt.
> >
> > | Metric   | wo /evidence | w /evidence  |
> > |----------|--------|----------------|
> > | P_ours   | 70.8   | 88.3           |
> > | P_LEMMA  | 78.8   | 90.4           |
> > | P_MMFB   | 75.0   | 87.5           |
> >
> >
> > As shown in the Table, without evidence, the accuracy is indeed sensitive to the prompt template. However, once we add our evidence strategy, the performance of all three prompts converges and becomes stable. These results indicate that while GPT-4o’s predictions are quite sensitive to prompt wording in the absence of evidence, but this sensitivity becomes almost negligible when strong evidence is provided. Therefore, the gains reported in Table 7 mainly come from our evidence-based design, rather than from prompt engineering alone. We will add this analysis and the detailed prompt templates to the revised paper.
> >
> > 4.**Insufficient Experimental Depth Due to Missing Fine-Tuning and Ablations for Multiple Pretrained Models**(Response to Weakness 4)
> >
> > Since we focus on accurate zero-shot evaluation and prioritize sourcing from real-world posts and provide accurate annotations, the size of our dataset cannot support supervised fine-tuning. The scarcity of reliable, expert-verified signals that meet real-world editorial standards makes it impractical to scale the dataset to a very large size.
> >
> > For the ablation analyses for multiple pretrained models, we are a bit unsure about what specific ablation analysis you refer to, and would greatly appreciate a brief clarification. Here, we added a more in-depth analysis on pretrained MLLMs (GPT, Gemini, Qwen, DeepSeek, LLaVA) to examine their performance across different error types when judging fake posts. This allows us to examine where the model struggles for detecting fake posts. We hope this addresses the issue, but please let us know if you were referring to a different type of ablation or additional experiments.
> >
> > As shown in the following Table, all models perform well in the Deepfake detection task. The relatively bad performance in Image OOC reflects the limitations of current MLLMs in identifying cross-modal misinformation, where the models need to establish semantic correlations between different domains and perform complex reasoning. The Text Misleading detection task requires models to perform complicated textual semantic reasoning between post caption and retrieved evidence, and models typically handle it better than Image OOC. This aligns with common observations that MLLMs are generally stronger in textual reasoning than in visual reasoning.
> >
> > | Model| Deepfakes (Acc.) | Image OOC (Acc.) | Text misleading (Acc.) |
> > |--|--|---|--|
> > | GPT-4o| 0.91| 0.89| 0.98|
> > | Deepseek-vl2| 0.87 | 0.87| 0.90 |
> > | Gemini-2.0-flash|0.83| 0.70| 0.72 |
> > | LLaVA-7b| 0.63| 0.47| 0.60 |
> > | Qwen-vl-7b| 0.55| 0.50| 0.36  |
> >
> > 5.**Unquantified Overlap in Multi-Label Posts** (Response to Weakness 5 and Question 6)
> >
> > Thank you for pointing this out. Regarding the statistics of multi-label posts and label overlap, we will add supplementary analyses in the revised version. First, we calculated the distribution of error-type labels across all fake posts and quantified the degree of label overlap. The results are shown in the following Table. As can be seen, approximately 7.8% of fake posts contain two or more error-type labels, and each fake post has 1.08 labels on average. The most frequent label combinations are 2 + 3 and 1 + 3, accounting for 5.3% and 2.3% of fake posts, respectively.
> > | Error Type | Count | % of Posts |
> > |---|--|--|
> > | Text Misleading | 247| 20.58% |
> > | Image Out-of-Context | 527| 43.92%|
> > | Deepfake  | 521| 43.42% |
> > | Text Misleading + Image Out-of-Context   | 64 | 5.33%|
> > |  Deepfake  + Text Misleading    | 28 | 2.33%|
> > | Deepfake + Image Out-of-Context   | 4 | 0.33%|

---

### Official Review · Reviewer_k1K4 · 2025-10-27

**Soundness:** 2
**Presentation:** 2
**Contribution:** 3
**Rating:** 2
**Confidence:** 3

**Summary:**

This work introduces the XFacta dataset, which evaluates the abilities of MLLMs to fact-check posts on social media containing both text and image. The dataset is collected from X/Twitter using community notes and reputable sources to categorize the posts into real and fake. The posts are then further categorized into "deepfakes", "image out of context", and "text misleading". An analysis of several fact-checking strategies is also presented.

**Strengths:**

1. Evaluating MLLMs on their ability to detect misinformation is an extremely important problem. Specifically, the multimodal aspect of misinformation is a challenging problem to solve and will require excellent multimodal reasoning from the MLLM.

2. The dataset presented is likely to be useful for practitioners seeking to construct AI systems for flagging misinformation online.

**Weaknesses:**

1. There are several other similar benchmarks for evaluating capabilities of MLLM to detect misinformation (cited in the paper itself). The authors argue that the dataset consists of contemporary posts unlike other benchmarks. In my opinion, this is insufficient because the presented benchmark may also get outdated soon in a couple of years. It would be better to construct a continuously evolving benchmark for this (similar to LiveBench/LiveCodeBench) .

2. It would also be good to include the URLs for the retrieved evidence (E_i, E_t) for each post in the dataset as this would enhance reproducibility + insulate the benchmark from future events that may change the veracity of the posts in question.

3. The evaluation of the different strategies for detecting misinformation is a bit perplexing. In particular, it is not clear why the strategies cannot be utilized simultaneously. For example, an MLLM agent may use start off using strategy 1, then if it's inconclusive it may move on to other strategies. Being constrained to use one strategy seems artificial. Also, it seems that the performance of the strategies is somewhat data dependent, it should be possible to construct datasets where a particular strategy is favored. I do not quite understand the usefulness of the insights obtained from this analysis.

**Questions:**

Please address the weakness above

---

> ### Author Response · Authors · 2025-11-22
>
> We appreciate the effort and time you have devoted to reviewing our paper. Below are our point-by-point responses to your comments.
>
> 1.**Limited Long-Term Relevance of the Benchmark** (Response to Weakness 1)
>
> Thank you for raising this important point. We agree that contemporary data alone cannot guarantee long-term relevance, and that any one-time snapshot of social media content will inevitably become outdated. Our emphasis on “contemporary posts” was not meant to suggest that this alone solves the problem, but rather to address a specific limitation of prior benchmarks: many existing datasets contain events that already appear in MLLM training corpora, introducing evaluation bias. Using recent posts helps mitigate this issue, but we fully agree that it is not sufficient to ensure longevity.
>
> To address the concern you raise, our benchmark is not intended to be static. As described in the paper, XFacta integrates a semi-automatic detection-in-the-loop pipeline. We first apply our detector to large streams of new social media posts, and human reviewers then verify the flagged items and continuously add newly confirmed real and fake posts to the dataset. This workflow enables XFacta to be updated regularly with minimal annotation cost, allowing the benchmark to evolve over time in a manner similar to LiveBench/LiveCodeBench, but tailored to misinformation detection, where fully automatic collection of false claims is still not feasible. We are also committed to regularly updating the dataset with our detector-in-the-loop pipeline to ensure that XFacta remains up to date and reflective of emerging real-world misinformation patterns.
>
> In addition, we will release all crawling scripts used for data collection, along with a maintained list of X accounts (which will also be updated regularly) belonging to professional fact-checking authors. This ensures transparency and allows other researchers to easily replicate our data collection pipeline, enabling continuous extension or reconstruction of the benchmark by the community.
>
> In summary, while contemporary posts help reduce memorization bias today, it is the continual update mechanism and the reproducible data collection pipeline that ensure XFacta functions as a living benchmark rather than a static one. We have revised the paper to clarify this motivation and design choice.
>
> 2.**Missing Evidence URLs Reduces Reproducibility** (Response to Weakness 2)
>
> In fact, we have already collected and stored the source URLs for all retrieved evidence associated with each post. We will include these URLs in the released dataset to ensure that future users can verify the evidence and fully reproduce our pipeline.
>
> 3.**Unnatural and Low-Insight Strategy Evaluation Design** (Response to Weakness 3)
>
> Thank you for the comment. In this section, we aim to analyze the contribution of each evidence type with as much clarity as possible. To achieve this, we begin by evaluating evidence sources individually, which allows us to clearly observe how each one influences the model’s behavior and which part of the detection process it primarily supports.
>
> When multiple evidence sources are used at the same time, their effects may interact in ways that make it difficult to understand which component is responsible for the improvement or degradation. By isolating evidence types, we can obtain more stable and controlled findings about how each source supports event verification or out-of-context detection. We also experimented with combined strategies (e.g., using all evidence types together, and using evidence extraction together with all evidence). As shown in the following table, these combined settings work as expected, and we will report these results in the supplementary material of the revised paper. However, they do not change the main insights obtained from the single-evidence analysis. Instead, the one-by-one setup provides the clearest view of how each retrieval method contributes to model behavior, which is the main goal of this part of the study.
>
> | Experiment Setup                   | Acc. | R.Acc. | F.Acc. |
> |------------------------------------|-------------|----------|----------|
> | All Evidence                       | 0.88        | 0.97     | 0.80     |
> | All Evidence + Extraction          | 0.91        | 0.98     | 0.84     |
> | T → Et + T → E + I → Et            | 0.75        | 0.81     | 0.69     |

---

> > ### Comment · Reviewer_k1K4 · 2025-11-25
> >
> > Thanks, I'm convinced by your response to W1 and W2, but I am still unsure of the usefulness of the analysis portion. I have increased my score to 4.

---

> > > ### Author Response · Authors · 2025-11-27
> > >
> > > We appreciate the improved rating and we understand the concern raised in weakness (3).
> > >
> > > To clarify, our understanding of your comment (3) is as follows:
> > >
> > > Different strategies may be useful for different subsets of the data. For example, some posts may require evidence type 1 to be resolved, others may favor evidence type 2, and so on. However, in our experiments, we apply the same strategy to all posts in a run and then compare aggregate accuracy across runs, rather than analyzing which specific posts benefit from which evidence type. If the data distribution shifted, the relative performance of the strategies could therefore change, which may limit the usefulness of our current analysis. We agree with this perspective, and we acknowledge that it can affect how broadly the insights should be used. Our response is:
> > >
> > > 1. **Goal of the analysis**.
> > >
> > >     Our main goal is not to argue that one strategy is globally “better” than another, but to understand what unique information each evidence type/strategy contributes to misinformation detection. For example, in Section 4.1.2, point 2, we state that T → Et (strategy 1) substantially boosts performance, especially for real posts. We then analyze why this strategy improves performance, what information it adds, and why the improvement is less significant for fake posts. We do not claim that this strategy dominates all others. The same style of analysis appears in points 5, 7, and 8. This isolated-strategy design allows us to obtain more stable and controlled findings, since we can attribute performance gains to specific evidence sources, whereas combined-strategy experiments, although more realistic, are inherently noisier and harder to interpret.
> > >
> > > 2. **“Better than” statements are case-specific**.
> > >
> > >     When we do compare strategies, the conclusions are explicitly conditioned on particular scenarios. For instance, in Section 4.1.2, point 3, we say that I → Et (strategy 3) is more effective than T → Ei (strategy 2) for out-of-context misinformation, and we manually inspect the success/failure cases to explain this behavior. We do not assert that I → Et is universally better than T → Ei for all types of data. Similarly, in point 6, we state that DuckDuckGo provides lower-quality evidence than Google, but this comparison is restricted to the same strategy instantiated with different search engines, and the “better” claim is about the search engine, not about the underlying strategy.
> > >
> > > 3. **Heterogeneity across misinformation types**.
> > >
> > >     Similar to your opinion, we also considered that different posts may favor different strategies. This is exactly the motivation for the experiment in Table 4, where we assume that different types of misinformation require different evidence types for effective detection. We indeed observe a trend that: Deepfakes are most effectively detected using the strategy I → Et, which achieves perfect accuracy (100%) in this category, while Image OOC and Text Misleading are better addressed by T → Ei and T → Et, respectively. This supports the view that strategy effectiveness is data- (and type-) dependent, and we will clarify this more explicitly in the paper text.
> > >
> > > 4. **New experiments with combined strategies**.
> > >
> > >     We also appreciate your suggestion that an MLLM agent could use multiple strategies sequentially or in combination, rather than being constrained to a single one. Based on your comment, we conduct an additional set of experiments that leverages multiple strategies simultaneously, beyond the “one-strategy-per-run” setting. Concretely, we explore:
> > >
> > >     (i) using all strategies together
> > >
> > >     (ii) selecting a non-redundant subset of strategies (e.g., using Google but not DuckDuckGo when both provide similar web search evidence)
> > >
> > >     (iii) allowing the MLLM to decide adaptively whether additional strategies/evidence are needed.
> > >
> > >     *(iii.a)  Parallel: We provide the input data and an explanation of each evidence type, then ask the MLLM to decide which evidence is needed. Then we use these selected pieces of evidence to assist the LLM in making decisions.*
> > >
> > >     *(iii.b) Sequential: Given the input data and one piece of evidence, the MLLM determines whether that evidence is conclusive enough to make a judgment. If not, we provide more pieces of evidence until a judgment can be made.*
> > >
> > > In these experiments, we utilize the GPT-4o model combined with Chain-of-Thought (COT) reasoning to guide the model's decision-making process. The results are:
> > >
> > > | |Acc.|R. Acc.|F. Acc.|
> > > |-|-|-|-|
> > > |(i) All Evidence|88.6|97.2|80.0|
> > > |(ii) non-redundant subset|90.8|97.5|84.2|
> > > |(iii.a) parallel|89.2|96.7|81.7|
> > > |(iii.b) sequential|85.0|85.8|84.2|

---

> > > > ### Author Response · Authors · 2025-11-27
> > > >
> > > > The experimental results show that when we use all evidence, the accuracy is higher than COT reasoning method (88.3%) in Table 5 of the paper, which uses I → Et and T → Et. When we use the non-redundant subset strategy, its performance outperforms the all-evidence strategy. This suggests that by combining more evidence, the model can indeed access more information, thereby improving its decision-making ability and accuracy. For the adaptive method (parallel), we found that the MLLM always outputs a fixed set of evidence (I → Et, T → Et, and Query → Et) and fails to fully utilize all available information. This behavior is similar to that observed in the sequential method, where the model typically stops after evaluating the first piece of evidence (I →Et). These phenomena suggest the current MLLMs exhibit a preference for specific pieces of evidence and don’t have an effective mechanism to balance and assess the relative importance of different pieces of evidence.

---

### Official Review · Reviewer_NcPq · 2025-10-31

**Soundness:** 2
**Presentation:** 2
**Contribution:** 2
**Rating:** 4
**Confidence:** 3

**Summary:**

This paper introduces XFACTA, a contemporary, real-world dataset for multimodal misinformation detection, composed of social media posts to avoid temporal bias and model memorization. XFACTA enables a rigorous analysis of MLLM-based detection methods, clarifying the relative impact of evidence retrieval versus reasoning. The study evaluates various MLLM architectures and existing approaches, providing key insights into effective design strategies. Furthermore, a semi-automatic detection-in-the-loop framework is proposed to continuously update the dataset, ensuring long-term relevance. XFACTA and the accompanying analysis serve as a robust benchmark for advancing multimodal misinformation detection research.

**Strengths:**

1 Introduces XFACTA, a novel dataset with real-world misinformation samples from X, avoiding temporal leakage and memorization bias common in outdated benchmarks.

2 Provides a clear disentanglement of the roles of evidence retrieval and reasoning in detection performance, offering actionable insights into model bottlenecks.

3 Proposes a semi-automatic detection-in-the-loop framework that continuously updates the dataset with new verified cases, ensuring long-term relevance and scalability.

**Weaknesses:**

1 While the dataset is curated with recent and real-world examples, its reliance on Twitter (X) as the primary source follows a well-established practice in prior work. The advancement over existing datasets is incremental.

2 The proposed semi-automatic detection-in-the-loop framework lacks a comprehensive methodological description. A clearer overview of the full detection pipeline, particularly with a schematic illustration or step-by-step workflow, would improve transparency and reproducibility.

3 The evaluation does not include direct comparison with human fact-checkers. As a result, it remains unclear how closely the system’s reasoning performance approaches human-level judgment in complex multimodal misinformation scenarios.

**Questions:**

1 Given that many existing multimodal misinformation datasets are also sourced from Twitter (X), what specific characteristics or design choices in XFACTA differentiate it from prior efforts, and how do these contribute to measurable improvements in evaluating MLLM-based detectors?

2 Could the authors provide a more detailed overview of the detection-in-the-loop pipeline, ideally with a schematic diagram or algorithmic description, to clarify the interaction between automated detection, human verification, and data update processes?

3 How does the system’s performance compare to that of human fact-checkers on the same multimodal misinformation tasks, particularly in cases requiring nuanced reasoning across text and image modalities?

---

> ### Author Response · Authors · 2025-11-22
>
> We appreciate the effort and time you have devoted to reviewing our paper. Below are our point-by-point responses to your comments.
>
> 1.**How XFacta Is Different from Prior Datasets and Contributes to Improvements in Evaluating MLLM-based Detectors** (Response to Weakness 1 and Question 1)
>
> **1 (a). Unique advantages of XFacta dataset compared with other datasets**
>
> In Table 1 of the main paper, we show systematic comparisons of our proposed XFacta and other existing datasets. Compared with previous work that uses Twitter, our XFacta dataset has several unique advantages:
>
> (1) Contemporary, avoiding overlap with MLLM training data.
>
> Earlier Twitter-based datasets were created before the widespread use of MLLMs, and thus are likely to overlap with MLLM training corpora, making zero-shot evaluation less reliable. In contrast, XFacta is collected from a strictly contemporary period, which prevents such data leakage and enables fair evaluation. (Line 208–210)
>
> To further address your concern, we conducted an additional experiment to show the importance of the contemporary feature. We selected GPT-4.1 as the base model for this experiment because its knowledge cutoff is June 2024. (We did not use GPT-4o since its cutoff is October 2023, while the earliest date in our dataset is January 2024, making it unsuitable). Based on the cutoff date, we divided the dataset into two subsets: 493 samples from January–June 2024 and 1907 samples from July 2024 onward. We used the CoT reasoning strategy for evaluation on the test set. For both subsets, we measured the detection accuracy with and without evidence.
>
> Without evidence, accuracy dropped from 81.8% → 73.3% when moving into the unseen period, confirming that out-of-cutoff misinformation is substantially harder for MLLMs. When external evidence was added, this gap largely disappeared (83.6% vs. 84.05%), indicating that GPT-4.1 relied heavily on memorized knowledge for the earlier subset.
>
> These findings quantitatively support our claim that contemporaneity is critical for fair evaluation: only a strictly contemporary dataset prevents shortcut solutions and forces MLLMs to perform real evidence-based reasoning.
>
> **January 2024 to June 2024**：
> | Condition| Acc.| R. Acc.| F. Acc.|
> |-|-|-|-|
> | wo/ evidence| 81.8| 81.8| 81.7    |
> | w/ evidence| 83.6| 84.8   | 82.4    |
>
> **July 2024 onward**：
> | Condition| Acc.| R. Acc.| F. Acc.|
> |-|-|-|-|
> | wo/ evidence| 73.3| 67.1| 82.4|
> | w/ evidence| 84.05| 83.59| 84.73|
>
> (2) Multimodal, capturing newly emerging misinformation patterns.
>
> The forms of multimodal misinformation on social media have evolved in recent years, and many older datasets do not reflect these updated patterns. XFacta is designed to capture these contemporary real-world multimodal signals, better matching today’s misinformation landscape. And we are also committed to regularly updating the dataset with our detector-assisted pipeline to ensure that XFacta remains up to date and reflective of emerging real-world misinformation patterns.
>
> (3) Evidence-based annotations, enabling reasoning evaluation.
>
> As described in the annotation procedure (Line 198–201), unlike datasets with weak or noisy labels, XFacta provides journalist-verified, evidence-grounded annotations. This supports not only the evaluation of prediction accuracy but also the assessment of model reasoning against journalist reasoning.
>
> (4) Moderate and balanced scale, suitable as a robust MLLM benchmark.
>
> While maintaining real-world authenticity, contemporariness, and high-quality evidence, XFacta offers a balanced scale (1,200 real and 1,200 fake examples). It is larger than most real-world datasets and more complete than recent contemporary datasets such as Post-4V (with 81 real and 105 fake examples), making it a reliable benchmark for MLLM evaluation.
> Taken together, though we acknowledge the substantial efforts behind prior misinformation datasets, XFacta offers unique advantages for the LLM era by aligning more closely with the challenges modern MLLMs actually face.
>
> **1 (b). XFacta contributes to measurable improvements in evaluating MLLM-based detectors**
>
> Beyond the advantages of the XFacta dataset itself summarized in 1(a), it also leads to measurable improvements in evaluating MLLM-based misinformation detectors.
>
> We summarize the key evaluation benefits below.
>
> (1) Stronger separation between “memorization” and evidence-based detection.
>
> (2) The dataset’s multimodal misinformation taxonomy (Deepfake, OOC, Text-Misleading) exposes which misinformation strategies each model fails on, something that earlier Twitter datasets can’t reveal.
>
> (3) XFacta eliminates topic and visual distribution biases during construction, producing fair evaluation and avoiding shortcuts.
>
> (4) Using XFacta, we compare retrieval variants and reasoning strategies in a controlled way and derive a concrete, reproducible recipe for stronger detectors. This turns XFacta into a benchmarking framework, not just a dataset.

---

> > ### Author Response · Authors · 2025-11-22
> >
> > 2.**Insufficient Description of the Detection Pipeline**  (Response to Weakness 2 and Question 2)
> >
> > Thank you for your suggestion. The core idea of our detection-in-the-loop pipeline is to let an automatic MLLM detector assign an initial label and reasoning, have human verifiers correct and refine the results, continuously enhance XFacta dataset with up-to-date posts, and then finally adapt and improve the detector iteratively.
> >
> > 1. **Detection-in-the-loop blueprint**
> >
> >     **Step 0: Data Preparation**
> >
> >     We first compile the set of posts to be evaluated by the detector.
> >
> >     **Step 1: Initial Automatic Detection**
> >
> >     An MLLM-based misinformation detector processes each post and outputs (i) a label (real / fake) and (ii) a rationale explaining the decision.
> >
> >     **Step 2: Human-Guided Verification**
> >
> >     Human reviewers check the correctness of the detector’s output, resolve ambiguous cases, and ensure that labels are reliable. In this way, the detection model reduces human workload while human verifications keep annotation quality high.
> >
> >     **Step 3: Dataset Update**
> >
> >     Only samples verified by humans are added to the dataset. This enables the dataset to grow with new events while remaining trustworthy.
> >
> >     **Step 4: Iteration and Dataset Augmentation**
> >
> >     Future researchers can apply the same strategies as in our paper for benchmarking to update their detectors with newer data. The updated detector is plugged back into Step 1 to re-label new or existing posts, and the same process is repeated. Across iterations, the verified set can be continuously merged into the XFacta dataset, so the detector and dataset are jointly improved with human-validated positive and negative signals.
> >
> > 2. **A small-scale experiment in our paper**
> >
> >     **Step 0: Preparing data**
> >
> >     We collect 500 new candidate posts from X/Twitter by monitoring accounts that frequently post trending or controversial content.
> >
> >     **Step1：Automatic detection**
> >
> >     All collected posts are processed by our detector. Among the 500 posts, the detector identifies 265 as fake and 235 as real, each accompanied by a reasoning trace.
> >
> >     **Step 2: Human verification**
> >
> >     We then manually examine the detector’s outputs. For each post, we check whether:
> >
> >     a. The detector’s reasoning is logically sound
> >
> >     b. There exists clear external evidence supporting the detector’s judgment.
> >
> >     If both conditions are satisfied, the post is accepted as a verified sample.
> >
> >     **Step 3: Dataset augmentation**
> >
> >     Following this verification process, 120 posts meet the above criteria and are added to our dataset as new high-quality entries. We will also release this augmented portion of the dataset publicly.
> >
> >     **Step 4: Implication for iterative updates**
> >
> >     The results show that our framework can reliably filter new real-world posts and add verified instances into XFACTA. This demonstrates that, as stronger LLMs become available, future researchers can reuse the same workflow: the improved detector will screen new data more accurately, and its outputs can again be human-verified and merged into the dataset.
> >
> >
> > Thus, the experiment provides that our framework can naturally support an iterative detection-in-the-loop process. We will revise the related part and add a schematic illustration based on your suggestions.
> >
> > 3.**Lack of Comparison with Human Fact-Checkers** (Response to Weakness 3 and Question 3)
> >
> > We appreciate your point about comparing our system with human fact-checkers. In our dataset, every misinformation post is accompanied by evidence and a full reasoning process written by professional journalists. These serve as high-quality human references. For the dataset release, we will include both the journalists’ evidence and reasoning, and the model’s full reasoning and judgment outputs. This will allow everybody to examine the model–human similarity in detail. And we will also include a model–human reasoning comparison as a case study in our revised paper.
> >
> > When it comes to comparisons with non-expert users (i.e., common users rather than trained journalists), which may be of particular interest, conducting such a study is currently impossible. We agree that comparing against non-expert humans would be valuable. However, fact-checking is not a simple multiple-choice task. It requires substantial time and real investigation, including searching for external evidence and verifying information from multiple sources. Running a controlled human study of this depth would demand significant effort and resources. We see this as an important future direction and will explore it in follow-up work.

---

> > > ### Comment · Reviewer_NcPq · 2025-11-26
> > > **I will keep my score**
> > >
> > > Thank you for the clarification. However, the biggest problem with this paper lies in the fact that the innovation in the data construction process is relatively minor. Although the author emphasized the advantages of the real-time nature of the dataset, this still remains questionable, as the updates and iterations of the current large models are also very rapid.

---

> > > > ### Author Response · Authors · 2025-11-27
> > > >
> > > > Thank you for the follow-up. We would like to clarify that XFacta is intended as a resource contribution, not as a methodological innovation in data construction. We do not claim novelty in the general pipeline of crawling and cleaning posts from X (Twitter). Instead, our contribution lies in (i) building a real-world dataset with carefully verified, high-quality labels, and (ii) maintaining its contemporality through a detection-in-the-loop updating pipeline.
> > > >
> > > > Regarding the concern that relying on X (Twitter) yields only incremental improvement: some prior datasets indeed also use this platform, but XFacta is designed to complement them rather than duplicate them. We focus on recent real-world content and apply strict cleaning and labeling procedures to ensure label correctness and reduce noise. This allows XFacta to better reflect the current distribution of multimodal misinformation, which existing static datasets may no longer capture well.
> > > > Regarding “real-time”  v.s. rapid model updates: XFacta’s collection and detection-in-the-loop pipeline are designed so that the dataset can be refreshed with new real-world examples, enabling evaluation under the current distribution rather than on historical data only. Our goal is therefore not to innovate on how data is constructed in the abstract, but to provide a robust, up-to-date real-world benchmark that better matches the practical deployment setting.
> > > >
> > > > We will revise the manuscript to more clearly position XFacta as a real-world, temporally updated benchmark (rather than a novel data-generation method) and to more explicitly differentiate it from prior X/Twitter-based datasets along these axes.

---

### Official Review · Reviewer_6Rdo · 2025-11-01

**Soundness:** 3
**Presentation:** 3
**Contribution:** 3
**Rating:** 6
**Confidence:** 2

**Summary:**

The paper introduces XFACTA, a contemporary, real-world multimodal misinformation dataset sourced from X/Twitter spanning Jan 2024–Apr 2025. Building on this dataset, the authors disentangle two core bottlenecks: evidence retrieval vs. reasoning. They design eight retrieval strategies plus two evidence post-processing steps (domain filtering, evidence extraction), and systematically compare multiple MLLMs across reasoning paradigms (CoT, prompt ensemble, self-consistency, multi-step). They also present a detection-in-the-loop data expansion pipeline and validate OOD generalization on timely Snopes cases.

**Strengths:**

1. The dataset’s contemporaneity (2024–2025) addresses memorization bias in current LLMs, ensuring fair evaluation beyond training knowledge leakage.
2. he authors use optimal transport alignment and topic matching to reduce distributional bias between real and fake samples, producing a credible and well-balanced corpus.
3. The dual-axis study (evidence retrieval vs. reasoning) provides a structured framework that clarifies where current MLLMs fail,  representing a crucial step toward principled detector design..

**Weaknesses:**

1. The process and consistency of manual review and evidence annotation are described briefly; inter-annotator agreement metrics and annotation protocols are missing.
2. The authors emphasize contemporaneity but do not systematically quantify how model performance degrades on older vs. newer misinformation, which would strengthen their central claim.
3. The annotations and the retrieved evidence may share the same sources (e.g., journalists’ debunking pages). Without strict separation between training and evaluation, this overlap can create shortcuts. The paper does not describe systematic deduplication and domain isolation procedures.
4. The paper aligns image distributions using optimal transport, but it lacks ablations with no alignment or alternative alignment methods to demonstrate the net contribution of this step to fairness.

**Questions:**

1. How was annotation quality controlled?  Were multiple annotators involved, and if so, what was the inter-annotator agreement (e.g., Cohen’s κ)?
2. Have you performed semantic deduplication across retweets/aggregator posts and near-duplicate image removal?  How do you ensure there is no leakage between Dev and Test?
3. How much fairness improvement is contributed by image distribution alignment (via optimal transport) ?  Have you conducted ablations or comparisons with alternative alignment methods ?

---

> ### Author Response · Authors · 2025-11-22
>
> We appreciate the positive evaluation of our paper. Below are our point-by-point responses to your suggestions.
>
> 1. **Missing Annotation Process Details** (Response to Weakness 1 and Question 1)
>
>     Thank you for pointing this out. Since journalists already give an authoritative judgment for the fake posts, and real posts come from official news accounts, our manual review mainly refers to data-level cleaning, such as deduplication, text and image denoising, confirming the authenticity of journalist accounts, and filtering out fake posts that lack sufficient evidence. Therefore, inter-annotator agreement is not applicable in this context. To improve transparency and reproducibility, we have added more details about the annotation protocol, including the guidelines and quality control steps. We hope this addresses your concern.
>
>
>     **More details of the annotation protocol**:
>
>     **Rule 1: Source Reliability**
>
>     Real data: Collected from authoritative news organizations (e.g., CNN, BBC, The Guardian).
>     Fake data: Collected from posts labeled as fake by professional journalists or community platforms (e.g., BBC fact-checking journalists, Helpful Community Notes).
>
>     **Rule 2: Evidence Completeness**
>
>     Real data: Must originate from verified news organizations.
>
>     Fake data: Must include at least one verifiable external piece of evidence demonstrating that the post contains misinformation. Posts without sufficient or reliable evidence will be discarded.
>
>     **Rule 3: Signal Quality**
>
>     Deduplication: Remove duplicated news posts to ensure uniqueness.
>
>     Text denoising: Remove corrupted characters, advertisements, and irrelevant URLs.
>
>     Image denoising: Discard low-quality images (e.g., blurred, overly compressed, or damaged).
>
>     **Rule 4: Logging**
>
>     Real data: Record the caption, image, and metadata.
>
>     Fake data: Record the caption, image, metadata, and the complete evidence chain supporting the fake label.

---

> > ### Author Response · Authors · 2025-11-22
> >
> > 2.**No Quantification of Temporal Performance Decay** (Response to Weakness 2)
> >
> > To strengthen the claims about contemporaneity effectiveness, following your suggestion, we conducted a cross-time experiment. We selected GPT-4.1 as the base model for our experiments because its knowledge cutoff is June 2024. (We did not use GPT-4o since its cutoff is October 2023, while the earliest date in our dataset is January 2024, making it unsuitable.) Based on the cutoff date, we divided the dataset into two subsets: 493 samples from January–June 2024 and 1907 samples from July 2024 onward. We used the CoT reasoning strategy for evaluation on the test set. For both subsets, we measured the detection accuracy with and without evidence.
> >
> > **January 2024 to June 2024**：
> > | Condition      | Acc. | R. Acc. | F. Acc. |
> > |----------------|------|---------|---------|
> > | wo/ evidence   | 81.8 | 81.8    | 81.7    |
> > | w/ evidence    | 83.6 | 84.8    | 82.4    |
> >
> > **July 2024 onward**：
> > | Condition      | Acc.  | R. Acc. | F. Acc. |
> > |----------------|-------|---------|---------|
> > | wo/ evidence   | 73.3  | 67.1    | 82.4    |
> > | w/ evidence    | 84.05 | 83.59   | 84.73   |
> >
> >
> >
> > The results directly quantify the effect of contemporaneity. We can see that the unseen portion (after July 2024) is noticeably harder for the model, showing a performance drop when misinformation falls outside the model’s knowledge period. When retrieved evidence is added, the performance gap narrows, indicating that GPT-4.1 indeed relies on memorization as a shortcut for misinformation detection.
> >
> > These findings support our central claim that the contemporary nature of our dataset is essential for fair evaluation of MLLMs' capability for misinformation detection.

---

> ### Author Response · Authors · 2025-11-22
>
> 3.**Potential Overlap Between Annotations and Retrieved Evidence** (Response to Weakness 3)
>
> Thank you for pointing this out. Indeed, if the annotations and the retrieved evidence come from the same sources, the model may exploit this overlap and develop shortcuts. To avoid this issue, all our experiments were conducted using the default domain filter, where we removed all domains that could overlap with the annotation sources. This ensures strict domain-level separation between the retrieved evidence and the annotation sources. We will add a detailed description of this filtering procedure in the paper to make it clearer.
>
>
> 4.**Lack of Ablations and Alternatives for Image Debiasing** (Response to Weakness 4 and Question 3)
>
> Thank you for pointing this out. We use the Optimal Transport (OT) to debias real vs fake images. To show its effectiveness, we conducted two ablation experiments to more systematically analyze the role of the alignment step in our method:
>
> (1) Feature-Level Alignment. If OT is effective, the feature distributions of real and fake images should become closer. To verify this, we use SigLIP as the feature extractor and compare cosine similarity between real and fake image features under two settings: (i) random selection, and (ii) images selected after applying OT.
>
> (2) LLM-Based Discriminability. If OT successfully reduces image bias, real and fake images should become harder to distinguish from pixels alone. In particular, an LLM that only sees the image should assign similar real (or fake) probabilities to images from real and fake posts; otherwise, there must be visual shortcuts that make real (or fake) posts easier to recognize. Therefore, we ask GPT-4o to classify whether images are from real vs. fake posts under two settings: (i) randomly selected images, and (ii) images selected after OT.
>
> | Setting           | Cosine similarity |
> |-------------------|-------------------|
> | Random Selection  | 0.19              |
> | OT Selection      | 0.38              |
>
> | Setting           | Images from real posts   | Images from fake posts   |
> |-------------------|---------------------------|---------------------------|
> | Random Selection  | 47% real, 53% fake        | 37% real, 63% fake        |
> | OT Selection      | 41% real, 59% fake        | 37% real, 63% fake        |
>
> From the experimental results, OT selection can better align real and fake image distributions, making them harder for the LLM to distinguish purely from visual content and preventing shortcutting based on image-only cues.
>
> We acknowledge that alternative debiasing strategies could also be applied, but we chose OT because it offers a robust, simple, and stable way to correct distributional imbalance. Comparing every possible alignment method isn’t the focus of our work, so we stick to the one that gets the job done reliably.
>
> 5.**Deduplication and Dataset Leakage Concerns** (Response to Question 2)
>
> After completing data collection, we performed a rigorous deduplication process to ensure that the dataset contained no duplicate or highly similar content. Each post includes a unique link, so we first conducted link-level deduplication and removed all samples sharing the same link, ensuring that no identical posts remained. After this step, we computed the semantic similarity of SBERT (all-MiniLM-L6-v2) text embeddings to identify posts with duplicate or highly similar content. Text pairs with similarity above a predefined threshold were removed, preventing repeated text or paraphrase-level near duplicates from appearing in the dataset. Regarding images, we applied perceptual-hash deduplication to ensure that no exact or near-duplicate visuals remained in the dataset. Specifically, we computed an average hash for each image and removed any sample whose Hamming-distance to an earlier image fell below a fixed cutoff. This procedure eliminates visually identical or highly similar images in a fully automatic and reproducible manner. In addition to this automated check, we also manually inspected all images and found only two to three visually similar pairs. Their proportion is extremely small and does not affect the statistical distribution of the dataset.

---

### Author Response · Authors · 2025-12-01
**AC Letter: Summary of Rebuttal & Discussion for Paper #8487**

Dear Area Chair,

Due to the recent system revert and the assignment of new ACs, we are writing to provide a concise summary of the consensus reached during the discussion period, as well as the specific remaining questions from the reviewers who have not yet replied.

**1. Score Update (4/2/4/6 → 4/4/4/6)**

Our initial scores were 4,2,4, and 6. During the discussion, **Reviewer k1K4 raised their score from 2 to 4** after we resolved a factual misunderstanding. Reviewer NcPq decided to keep the score. The other two reviewers did not reply before the system revert.

**2. Resolution with Reviewer vHFL (Score raised 2 → 4)**

The reviewer’s main concerns are that (1) the benchmark will soon become outdated. (2) adding evidence URLs to improve reproducibility.  (3) why we evaluated each strategy separately rather than trying to use them simultaneously.

- **Resolution**: we have (1) clarified that XFacta is designed as a continuously updated benchmark supported by a detector-in-the-loop pipeline, and (2) clarified that all evidence URLs have been collected and will be released to enhance reproducibility.

- **Outcome**: The reviewer acknowledged that the misunderstanding about concerns (1) and (2) was resolved and raised their score.

- **Remaining Issue**: The reviewer remained unconvinced about (3) the usefulness of evaluating the strategies separately.

- **The Explanation**: We explained that the reason we isolate each evidence type is to **measure its individual contribution and obtain more stable and controlled findings.** We also added combined-strategy experiments and found that integrating multiple strategies leads to more robust performance. But for the analysis purpose, it is sufficient to evaluate strategies separately.

- **Final note**: The reviewer did not have the chance to reply before the discussion closed.

**3. Resolution with Reviewer NcPq (Score raised 4 → 4)**

The reviewer’s concerns are that (1) the improvements of our dataset are only incremental, (2) the detection-in-the-loop pipeline is not described clearly enough, and (3) the evaluation lacks comparison with human fact-checkers.

- **Resolution**: We clarified that (1) XFacta offers meaningful improvements over prior Twitter-based datasets through strict contemporaneity, multimodal coverage, and evidence-based annotations, and we provided new experiments supporting this. We also supplied (2) a clear step-by-step description of the detection-in-the-loop pipeline and will add a schematic illustration. Finally, we explained that (3) journalist-verified reasoning already enables model–human comparison, while broader human studies are valuable but currently impractical.

- **Outcome**: The concerns of (2) and (3) are addressed. But the reviewer still believes that (1) the data construction process lacks novelty and notes that its real-time advantage will diminish as time goes on.

- **The Explanation**: XFacta is intended as a **resource contribution**, not as a methodological innovation in data construction. **We never positioned data-construction novelty as our contribution**. Our contribution lies in providing carefully verified, contemporary multimodal misinformation and maintaining this contemporality through a detection-in-the-loop updating pipeline. Although it also draws from X (Twitter), XFacta complements prior datasets by offering cleaner labels and an up-to-date distribution that static datasets no longer capture.

- **Final note**: The reviewer did not have the chance to reply before the discussion closed.

**4. Reviewers 6Rdo (Rating 6) and WoLQ (Rating 4)**

These two reviewers did not respond during the discussion period. Despite the difference in scores, their views can be summarized as follows.

- **The Concerns**:

    (1) unclear dataset labeling details and inter-annotator agreement metrics

    (2) missing experiment (6Rdo asked to quantify contemporary nature and add an ablation experiment about optimal transport alignment, WoLQ asked to add a prompt-sensitivity experiment and pretrained model ablation)

    (3) small misunderstandings that need clarification

- **The Explanation**: We have (1) clarified the labeling procedure, and (3) addressed the confusion and misunderstandings. We have also (2) added all the requested ablation studies accordingly.

- **Final note**: We believe that both reviewers can resolve their concerns based on our clarifications and new experimental results.

Thank you for your time and for managing this challenging situation.

Best regards,

Authors of Paper #8487

---

### Meta-Review · Area_Chair_jeU6 · 2026-01-05

**Summary:**

Reviewers acknowledged the dataset's contemporaneity but questioned its technical novelty, small scale, and the lack of human-level benchmarks. Concerns also persisted regarding the labeling pipeline's transparency and the long-term utility of  dataset despite the proposed update pipeline.

**Reviewer Concerns:**

The rebuttal successfully quantified temporal decay, validated image debiasing, and proved results are not purely prompt-dependent.
Concerns regarding incremental innovation for Twitter-based datasets and the absence of human fact-checker comparisons remain. Reviewers still view the sample size as insufficient for broad diversity.

**Reviewer Scores:**

6Rdo (6): Likely stays at 6 but with lower confidence due to unresolved concerns.

k1K4 (4): Already increased from 2 to 4.

NcPq & WoLQ (4): Likely stay at 4, as they prioritize methodological innovation.

---

### Decision · Program_Chairs · 2026-01-26

Reject